# Adaptive Fourier Decomposition-guided Neural Operator Design for Inverse PDE Problems

## Abstract

Inverse problems, which are generally ill-posed, aim to identify the unknown parameters of a physical system from the observations of its output. A large class of inverse problems for partial differential equations (PDEs) are only well-defined as mappings from operators to functions. However, existing operator learning frameworks either do not explicitly account for the underlying operator space or solve the inverse problems in a Hilbert space. Meanwhile, it has been shown that a Banach space setting for the parameter space would be closer to reality for a wide range of problems. Driven by this, we introduce AFDONet-inv, a novel neural operator solver whose design is rigorously guided by adaptive Fourier decomposition (AFD) theory, to solve inverse problems for PDEs in a Banach space. Each component of AFDONet-inv's architecture, including primal and dual nets, latent-to-RKBS (reproducing kernel Banach space) network, and dynamic convolutional kernel network (CKN), has a corresponding component in the AFD operation in Banach space. This way, AFDONet-inv is mathematically explainable and grounded in the AFD theory and possesses several desirable properties. Extensive experiments demonstrate that AFDONet-inv outperforms state-of-the-art inverse PDE solvers in terms of solution accuracy.

## 1 Introduction

A wide range of scientific and engineering systems and phenomena, including fluid dynamics, heat and mass transport, structural mechanics, and cell growth, can be characterized and modeled by partial differential equations (PDEs). A fundamental task in understanding the dynamics of these systems and phenomena is to accurately infer unknown parameters $\alpha$ in the PDE from observed solution data, which might be noisy. Unlike many PDE forward problems, inverse problems are generally ill-posed, in the sense that their solutions either do not exist, are not unique, or are unstable to small perturbations in the data (Kirsch, 2021). For example, the inverse heat equation, which attempts to infer a previous temperature profile based on the final temperature distribution, is ill-posed in that the solution is highly sensitive to changes in the final data. Ill-posed PDE inverse problems are considerably harder than their forward problems.

Traditionally, PDE inverse problems are solved via PDE-constrained optimization (Hinze et al., 2009) or Bayesian inference (Stuart, 2010). In PDE-constrained optimization framework, the goal is to minimize the difference between the observed data and the PDE solution, and the PDE is enforced as a constraint. The resulting optimization problem can be solved using adjoint methods or Tikhonov regularization. Meanwhile, in the Bayesian inference framework, the inverse problem is formulated as a statistical inference problem. And the goal is to estimate the posterior distribution of PDE parameters given the solution data. This usually requires sampling over the parameter space and solving the forward PDE multiple times. These conventional methods require repeated solutions of forward PDEs and thus often suffer from high computational cost, especially for nonlinear or high-dimensional problems.

Data-driven solutions of PDEs have emerged as a powerful paradigm to address these challenges. Among them, two of the most popular methodological advancements are physics-informed neural network (PINN) and operator learning. PINN-based methods directly embed physical laws into the

loss function or neural network architecture (Raissi et al., 2017; 2019). By minimizing the total loss, PINN can effectively solve the PDE, fit the data, and infer the parameters simultaneously. PINN-based solvers have been applied to a wide range of inverse problems, including ocean engineering (Jagtap et al., 2022b), supersonic flow (Jagtap et al., 2022a), nano-optics and metamaterial design (Chen et al., 2020b), to name a few. However, in these methods, the PDE is enforced as a soft constraint and thus faces the intrinsic trade-off between fitting the data and solving the PDE accurately. Furthermore, many PINN-based methods focus on solving a single problem instance, which poses computational efficiency challenges when it comes to real-time inference to multiple instances (Cho & Son, 2025).

Meanwhile, operator learning, which aims to directly learn the mapping between infinite-dimensional function spaces (e.g., from input functions to solutions), can enable fast, mesh-independent approximation of PDE solutions across different input conditions. Existing operator learning approaches for solving forward PDE problems include the deep operator network (Deep-ONet) (Lu et al., 2019; 2021) and its variants, as well as neural operator (Kovachki et al., 2023) and its variants, including Fourier neural operator (FNO) (Li et al., 2020a; 2023), wavelet neural operator (WNO) (Tripura & Chakraborty, 2023), multiwavelet neural operator (MWT) (Gupta et al., 2021), etc.

Following these breakthroughs, various operator learning methods have been developed to solve inverse PDE problems. For instance, Wang & Wang (2024) presented a latent neural operator (LNO) method to solve forward and inverse PDE problems in their latent space. Molinaro et al. (2023) proposed neural inverse operator (NIO), which integrates DeepONet and FNO, to solve inverse PDE problems. More recently, Cho & Son (2025) developed a physics-informed operator learning framework called PI-DION for inverse problems without the need for labeled training data. Yu et al. (2024) introduced a new neural operator architecture called nonlocal attention operator (NAO) based on the attention mechanism and showed that the attention mechanism is equivalent to a double integral operator that enables nonlocal interactions among spatial tokens. This allows NAO to address ill-posedness and rank deficiency in inverse PDE problems. Building upon the NAO framework, Liu & Yu (2025) developed a scalable attention-based neural operator architecture called neural interpretable PDEs (NIPS). Using linear attention reformulation and Fourier convolution, NIPS achieved enhanced predictive accuracy and computational efficiency compared to NAO.

Despite these advancements, existing operator learning frameworks for inverse PDE problems face two critical drawbacks. First, existing frameworks either do not explicitly account for the underlying operator space or solve the inverse problems in a Hilbert space. Meanwhile, it has been shown that a Banach space setting for the parameter space would be closer to reality for a wide range of problems. However, this is an important aspect that previously have been overlooked. Second, the design of exact neural architectures in many existing inverse PDE solvers has been "more of an art than a science" (Sanderse et al., 2025), requiring significant intuition, expert experience, and trial-and-error experimentation. And rigorous mathematical basis and explainability have been lacking in the design. To address both challenges, here we introduce AFDONet-inv, a novel neural operator for solving inverse PDE problems in Banach space, whose design is systematically guided by the adaptive Fourier decomposition (AFD) theory. Each component in the neural architecture, including primal and dual nets, latent-to-RKBS (reproducing kernel Banach space) network, and dynamic convolutional kernel network (CKN), has a corresponding component in the AFD operation. This way, AFDONet-inv is mathematically explainable and grounded in the AFD theory, thereby providing rigorous theoretical justifications and performance guarantee. Through extensive experiments, we demonstrate that AFDONet-inv outperforms state-of-the-art inverse PDE problem solvers in terms of solution accuracy.

## 2 PRELIMINARIES

### 2.1 INVERSE PROBLEM IN HILBERT VS. BANACH SPACES

A Hilbert space is a complete inner product space (i.e., a vector space equipped with an inner product that induces a norm), and every Cauchy sequence in the space converges with respect to this norm. The interpretability given by the inner product has enabled rigorous convergence analysis and comprehensive application regularization techniques to be applied for solving inverse problems in Hilbert spaces over the last decades. However, for numerous inverse problems in PDEs, the reasons

for using a Hilbert space setting seem to be based on conventions rather than an appropriate and realistic model choice. In fact, it has been shown that the nature of Hilbert spaces cannot accurately capture the structures of parameter space for many PDEs, and often a Banach space setting would be closer to reality (Schuster et al., 2012). As a generalization of the Hilbert space, a Banach space is a complete normed vector space (i.e., a vector space equipped with a norm such that every Cauchy sequence in the space converges with respect to this norm), and the main difference between Hilbert and Banach spaces is the existence of an inner product and thus orthogonality. Banach spaces are more suitable for solving inverse problems involving sparsity, discontinuities, or measure-valued representations.

## 2.2 ADAPTIVE FOURIER DECOMPOSITION (AFD)

Adaptive Fourier decomposition (AFD) is a novel signal decomposition technique, which is essentially established as a new approximation theorem in a reproducing kernel Hilbert space (RKHS) sparsely in a given domain $\Omega$ as $\sum_{i=1}^{\infty} \langle s, \mathscr{B}_i \rangle \mathscr{B}_i$ for the chosen orthonormal bases $\mathscr{B}_i$ (Qian, 2010; Qian et al., 2012; Saitoh et al., 2016). Compared to conventional signal decomposition approaches, AFD achieves higher accuracy and significant computational speedup. AFD was first proposed for the Hardy space (Qian, 2010; Qian et al., 2011; 2012), then extended to the Bergman space (Wu et al., 2022), random signals (Qian, 2022), and manifolds (Song & Sun, 2022).

For classic AFD in RKHS, the sparse bases $\{\mathscr{B}_i\}_i$ are made orthonormal to each other by applying the Gram-Schmidt orthogonalization process to the normalized reproducing kernels associated with different "**poles**", which are a set of complex numbers $\{a_i\}_i$ that are adaptively selected. For instance, in classic AFD in Hardy space $H^2$ (a specific type of Hilbert space consisting of holomorphic functions defined on the unit disk), a common choice of reproducing kernel is the normalized Szegö kernel, defined as $e_a(z) = \frac{\sqrt{1-|a|^2}}{1-\bar{a}z}$, where $a$ belongs to the unit disk. Then, to construct the orthonormal bases $\mathscr{B}_i$, one selects a sequence of distinct poles $\{a_i\}_i$, substitutes them into the normalized Szegö kernel expression, and applies the Gram-Schmidt orthogonalization process on $e_{a_i}(z)$.

To adaptively select the sequence of poles such that convergence of AFD approximation is ensured, one shall follow the so-called "**maximal selection principle**" (Song & Sun, 2022), such that the resulting $|\langle s, \mathscr{B}_i \rangle|$ is as large as possible. This is similar to a greedy search algorithm. Specifically, to select the next pole $a_i$ given $i-1$ already selected poles, $a_1, \ldots, a_{i-1}$ (hence bases $\mathscr{B}_1, \ldots, \mathscr{B}_{i-1}$), the corresponding orthonormal basis $\mathscr{B}_i$ needs to satisfy:

$$|\langle s, \mathscr{B}_i \rangle| \geq \rho_i \sup \left\{ \langle s, \mathscr{B}_i^{b_i} \rangle \, | b_i \in \Omega \backslash \{a_1, \ldots, a_{i-1}\} \right\}, \tag{1}$$

where $0 < \rho_0 \leq \rho_i < 1$, $\mathscr{B}_1^{b_1} = \frac{k_{b_1}}{\|k_{b_1}\|_{H(\Omega)}}$ and $\mathscr{B}_i^{b_i} = \frac{k_{b_i} - \sum_{j=1}^{i-1} \langle k_{b_i}, \mathscr{B}_j \rangle \mathscr{B}_j}{\|k_{b_i} - \sum_{j=1}^{i-1} \langle k_{b_i}, \mathscr{B}_j \rangle \mathscr{B}_j\|_{H(\Omega)}}$. Here, $k_{b_i}$ is the reproducing kernel at $b_i$. Under the under maximal selection principle, the convergence of AFD approximation has been proven in Song & Sun (2022). In actual implementation, however, we often want to strengthen Equation equation 1 by introducing an extra **bias term** denoted as $\gamma_i > 0$ to enhance convergence. Essentially, this ensures that $|\langle s, \mathscr{B}_i \rangle|$ is always greater than the RHS of Equation equation 1 by at least $\gamma_i$. And the resulting updated maximal selection principle formulation becomes:

$$\gamma_i \leq |\langle s, \mathscr{B}_i \rangle| - \rho_i \sup \left\{ \langle s, \mathscr{B}_i^{b_i} \rangle \, | b_i \in \Omega \backslash \{a_1, \ldots, a_{i-1}\} \right\}. \tag{2}$$

## 3 AFD IN REPRODUCING KERNEL BANACH SPACE (RKBS)

The classic AFD, which operates on RKHS, leverages orthonormal bases. On the other hand, one cannot properly define orthogonality and inner product in a Banach space. Instead, we extend the inner product definition by adopting the concept of "**dual pairing**", denoted as $\langle \cdot, \cdot \rangle_{\mathcal{B}, \mathcal{B}^*}$ for the primal RKBS $\mathcal{B}$ and its dual $\mathcal{B}^*$. Essentially, a dual pairing is a non-degenerate bilinear map between two vector spaces that produces a scalar (Brezis & Brézis, 2011). With this, we will adaptively identify poles following a similar maximal selection principle, such that $|\langle r_{i-1}, J(\mathscr{B}_i) \rangle_{\mathcal{B}, \mathcal{B}^*}|$ is as large as possible. Here, $r_{i-1} = s - s_{i-1}$ is the **residual** between the true signal $s$ and its $(i-1)$-th decomposed component, $s_{i-1}$. And $J$ is called **duality map**, which satisfies $J(\mathscr{B}_i) \in \mathcal{B}^*$ and

$\langle \mathscr{B}_i, J(\mathscr{B}_i) \rangle_{\mathcal{B}, \mathcal{B}^*} = ||\mathscr{B}_i||_{\mathcal{B}}^2$. Specifically, to select the next pole $a_i$, the corresponding basis $\mathscr{B}_i$ needs to satisfy:

$$\gamma_i \leq |\langle r_{i-1}, J(\mathscr{B}_i) \rangle_{\mathcal{B}, \mathcal{B}^*}| - \rho_i \sup_{\mathscr{B}_i \in \mathcal{D}} \{|\langle r_{i-1}, J(\mathscr{B}_i) \rangle_{\mathcal{B}, \mathcal{B}^*}|\}, \tag{3}$$

which is analogous to Equation equation 2. In actual implementation, we find that setting $\gamma_i$ to a fixed value of $0.5$ works well for most problem settings. Here, we define set $\mathcal{D} = \{\mathscr{B}_i | a_i \in \mathcal{B}\}$ (note that we no longer need to exclude already selected poles as the concept of orthogonality does not hold in RKBS anymore), $\gamma_i$ is the bias term, and $0 < \rho_0 \leq \rho_i < 1$. With this, the AFD operations in RKBS give $s = \sum_{i=1}^{\infty} \langle r_{i-1}, J(\mathscr{B}_i) \rangle_{\mathcal{B}, \mathcal{B}^*} \mathscr{B}_i$. We remark that, to the best of our knowledge, such generalization of the AFD theory to Banach space has not been proposed before. The technical details and mathematical properties of AFD in RKBS are discussed in Appendix A.

# 4 AFD-GUIDED NEURAL OPERATOR DESIGN

Architecture overview

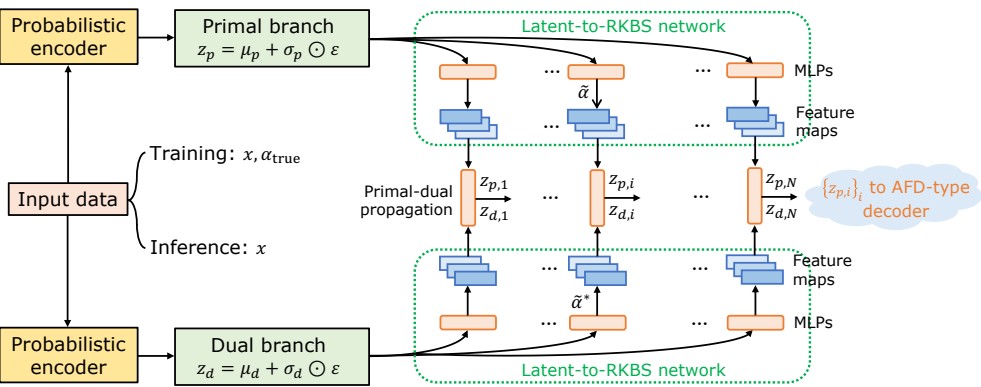

AFD-type dynamic CKN decoder

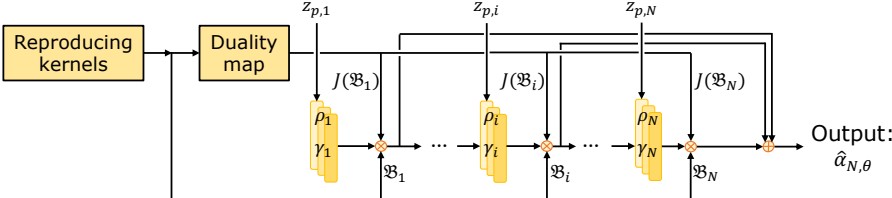

Figure 1: Our proposed AFDONet-inv framework, whose design is guided by the AFD theory and operation, for solving inverse PDE problems in Banach space. Note that the elements in the figure are static representations of the corresponding theoretical component, whereas the actual computation follows the dynamic, recursive update defined by Equation equation 12.

Once we establish the theoretical framework for AFD in Banach space, we design a tailored neural operator architecture, which we name as AFDONet-inv, that reproduces and realizes this theoretical framework. AFDONet-inv is an AFD-based VAE architecture (see Figure 1) to solve inverse PDE problems in Banach space. After the encoder, AFDONet-inv identifies the closest RKBS where the latent variables reside using a latent-to-RKBS network. Subsequently, AFDONet-inv reconstructs the PDE parameters by adaptively selecting the poles in a specially designed decoder network, thereby resembling the AFD operation.

## 4.1 NEURAL ARCHITECTURE

**The encoder network.** The encoder network maps the input $x \in \mathbb{R}^d$ (e.g., PDE solutions in training dataset during training or test dataset during inference stage) to latent variables in both the primal

and dual latent spaces, which correspond to a Banach space $B$ and its dual space $B^*$, respectively. Note that during training, since the PDE parameters $\alpha_{\text{true}}$ only appear in the loss function, they are part of training dataset but not part of the input $x$ to the encoder. Each encoding branch follows the standard VAE framework. That is, for the primal branch:

$$\left(\mu_p(x),\, \log \sigma_p^2(x)\right) = W_{p,2}\left(\phi\left(W_{p,1}x\right)\right), \tag{4}$$

$$z_p = \mu_p(x) + \sigma_p(x) \odot \varepsilon, \quad \varepsilon \sim \mathcal{N}(0, I), \tag{5}$$

where $W_{p,1} \in \mathbb{R}^{w \times d}$, $W_{p,2} \in \mathbb{R}^{2r \times w}$ are the weight matrices, $\phi(\cdot)$ is the activation function, and $z_p \in \mathbb{R}^r$ is the latent variable in the primal space.

Similarly, for the dual branch:

$$\left(\mu_d(x),\, \log \sigma_d^2(x)\right) = W_{d,2}\left(\phi\left(W_{d,1}x\right)\right), \tag{6}$$

$$z_d = \mu_d(x) + \sigma_d(x) \odot \varepsilon, \quad \varepsilon \sim \mathcal{N}(0, I), \tag{7}$$

where $W_{d,1} \in \mathbb{R}^{W_e \times d}$ and $W_{d,2} \in \mathbb{R}^{2r \times W_e}$ are the dual encoder weight matrices, and $z_d \in \mathbb{R}^r$ is the dual latent variable in $B^*$.

**The latent-to-RKBS network.** Given the latent variables $z_p$ and $z_d$, our goal is to determine their values in the parameter space, which lies in a RKBS. To do this, we extend the latent-to-kernel idea from Lu et al. (2020) and design a latent-to-RKBS network to project the latent variable $z_p$ to its nearest RKBS where the kernel is constructed. And then, its dual space in which the latent variable $z_d$ lie is obtained via the duality map $J(v) = \|v\|_{\mathfrak{p}}^{\mathfrak{p}-2} \cdot |v|^{\mathfrak{p}-2} \cdot v$ in the form of $L^{\mathfrak{p}}$ norm. First, $z_p$ and $z_d$ are respectively mapped to the parameter space $\tilde{\alpha} \in B$ and $\tilde{\alpha}^* \in B^*$ through two identical multilayer perceptron networks (MLPs). Then, feature maps $\text{FM}(\cdot)$ will project $\tilde{\alpha}$ and $\tilde{\alpha}^*$ onto their corresponding RKBS $\mathcal{B}$ and its dual $\mathcal{B}^*$, respectively. In other words, the feature map network is designed to map the latent vector $z_p \in \mathbb{R}^r$ to $N$ scalar coefficients in $\mathbb{R}$ needed for the kernel decomposition (i.e., $\text{FM} : \mathbb{R}^r \to \mathbb{R}^N$). The latent-to-RKBS network learns the feature maps from a Banach space $B$ to its nearest RKBS $\mathcal{B}$, where the reproducing kernel $K_z$ for any pair of encoder inputs (i.e., PDE solutions in training or test dataset) is given by:

$$K_z(x_1, x_2) = \sum_{i=1}^{N} \text{FM}_i(z_p) \cdot k_i(x_1, x_2). \tag{8}$$

In Equation 8, $N$ is the total number of basis kernels, $\text{FM}_i(z_p)$ is the $i$-th feature map coefficient for latent variable $z_p$, and $\{k_i(x_1, x_2)\}_i$ are the learned basis kernels are based on a Fourier spectral kernel formulation. Here, $k_i(x_1, x_2)$ contains a series of learned parameters in the spectral domain. Since these basis kernels are in RKBS, which is closed under finite linear combinations, $K_z$ lies in the RKBS as well.

**Primal-dual propagation.** The concept of dual pairing is realized in a primal-dual propagation network. Consisting of a primal net and a dual net, it propagates and refines the feature representations $\text{FM}(\tilde{\alpha}) \in \mathcal{B}$ and $J\left(\text{FM}(\tilde{\alpha})\right) \in \mathcal{B}^*$, which correspond to the latent variables $z_p$ and $z_d$, respectively. Here, the primal net performs the spectral convolution $\mathcal{S}(f)(x)$, which transforms a given function $f$ into the frequency domain via a 2D Fourier transform (Li et al., 2020a):

$$\mathcal{S}(f)(x) = \mathcal{F}^{-1}\left[\chi(\xi) \cdot \mathcal{F}(f)(\xi) \cdot W(\xi)\right](x), \tag{9}$$

where $\mathcal{F}[\cdot](\xi)$ is the Fourier transform at $\xi$, $\mathcal{F}^{-1}[\cdot](x)$ is its inverse Fourier transform at $x$, $\chi(\xi)$ denotes the mode selector, and $W(\xi)$ refers to the learnable weights in the frequency domain. From an RKBS perspective, the spectral convolution $\mathcal{S}(f)(x)$ is equivalent to a nonlocal kernel operator $\sum_{\xi} w_{\xi} \cdot \langle f, \varphi_{\xi} \rangle \cdot \psi_{\xi}(x)$, where $w_{\xi}$ are learnable weights in the frequency domain, $\varphi_{\xi} \in \mathcal{B}^*$ is the basis in the dual space, and $\psi_{\xi} \in \mathcal{B}$ is the biorthogonal primal basis which satisfies $\langle \psi_{\xi}, \varphi_{\eta} \rangle = \delta_{\xi\eta} = 1$ if $\xi = \eta$ and $= 0$ otherwise (Zhang et al., 2009). On the other hand, the dual net performs a point-wise convolution $\mathcal{C}(f)(x)$ as (Hua et al., 2018):

$$\mathcal{C}(f)(x) = \sum_{c'=1}^{C} W_{cc'} f^{(c')}(x), \tag{10}$$

where $C$ is the number of input channels of $f$, $f^{(c')}(x)$ refers to the value of $f$ at $x$ and channel $c'$, and $W_{cc'}$ denotes the learned map from channel $c'$ to channel $c$. This point-wise convolution resembles a Dirac-type kernel integral on its domain $\Omega$ as:

$$\mathcal{C}(f)(x) = \int_\Omega \delta(x - y) \cdot W \cdot f(y) \mathrm{d}y. \tag{11}$$

Finally, the residuals in the primal space and its dual space after each layer are updated and propagated to the corresponding latent variables as:

$$z_{p,i} = \mathrm{GELU} \circ \mathrm{BN}\left(z_{p,i-1} + \mathcal{S}(z_{p,i-1}) + \mathcal{C}(z_{p,i-1})\right),$$
$$z_{d,i} = z_{d,i-1} + \mathcal{S}(z_{d,i-1}) + \mathcal{C}(z_{d,i-1}), \tag{12}$$

where $z_{p,i}, z_{d,i}$ are the residuals of $i$-th layer in the primal space and its dual space with $z_{p,0} = \mathrm{FM}(\tilde{\alpha})$ and $z_{d,0} = J(\mathrm{FM}(\tilde{\alpha}))$, respectively. The initial latent vector, $z_p$, is used to define $\tilde{\alpha}$, and $\tilde{\alpha}$ then defines the initial decoder input, $z_{p,0}$. Here, GELU denotes the GELU activation function and BN is the batch normalizing transform (Ioffe & Szegedy, 2015).

It can be shown that $z_p$ and $z_d$ in Equation equation 12 actually correspond to the residual $r_i$ and its dual $J(r_i)$ defined in the AFD theory as:

**Theorem 4.1.** *Let $\mathcal{B}$ be a uniformly smooth and uniformly convex Banach space, $r_i \in \mathcal{B}$ be the residual at step $i$ in the AFD process, $z_p$ and $z_d$ are defined in Equation equation 12. Then, for any $\varepsilon > 0$, there exists a choice of parameters $W(\xi)$, $W_{cc'}$, and BN scalars such that:*

$$\|z_{p,i} - r_i\|_{\mathcal{B}} < \varepsilon, \text{and} \ \|z_{d,i} - J(r_i)\|_{\mathcal{B}^*} < \varepsilon.$$

*Proof.* See Appendix B. □

Note that, even though the variables $z_{p,i}$ are computationally represented as tensors, Theorem 4.1 shows that the learned network outputs behave as if they were the true theoretical Banach space residuals $r_i$.

**The AFD-type decoder network.** Once the RKBS and its reproducing kernel $K_z$ are constructed, we design a decoder network based on the AFD operation to reconstruct parameters $\alpha$ from $z_{p,i}$. First, we normalize the reproducing kernel $K_z$ in Equation equation 8 as $\mathcal{B}_i(\cdot) = \frac{K_z(\cdot, a_i)}{\|K_z(\cdot, a_i)\|_{\mathcal{B}}}$, each associated with a pole $a_i$. The set of these normalized reproducing kernels is denoted as $\mathcal{D} = \{\mathcal{B}_i | a_i \in \mathcal{B}\}$. The decoder then adopts a dynamic convolutional kernel network (CKN) (Mairal et al., 2014; Chen et al., 2020a), in which, for each convolutional layer $i$, (i) performs dual pairing between $z_{p,i}$ and the normalized reproducing kernel $\mathcal{B}_i$, (ii) assigns a multiplier $0 < \rho_0 \leq \rho_i < 1$ to the output of each convolutional layer, and (iii) incorporates skip connections for each convolutional layer. Finally, the output of the dynamic CKN containing $N$ convolutional layers is:

$$\hat{\alpha}_{N,\theta} = \sum_{i=1}^{N} \langle z_{p,i}, J(\mathcal{B}_i)\rangle_{\mathcal{B},\mathcal{B}^*} \mathcal{B}_i. \tag{13}$$

Guided by the AFD theory, the selection of poles and their reproducing kernels follows a similar maximal selection principle as in the AFD theory. Here, starting from Equation equation 3, we can write an analogous condition for selecting poles as:

$$\gamma_i \leq |\langle z_{p,i-1}, J(\mathcal{B}_i)\rangle_{\mathcal{B},\mathcal{B}^*}| - \rho_i \sup_{\mathcal{B}_i \in \mathcal{D}} \{|\langle z_{p,i-1}, J(\mathcal{B}_i)\rangle_{\mathcal{B},\mathcal{B}^*}|\}. \tag{14}$$

With this, and leveraging the convergence behavior of AFD, we can show that our decoder in AFDONet-inv converges as $N \to \infty$ by the following theorem:

**Theorem 4.2.** *Let $\mathcal{B}$ be a uniformly smooth and uniformly convex Banach space, and let $\hat{\alpha}_{N,\theta}$ be the output of the dynamic CKN decoder with $N$ layers. By selecting poles and bias terms following the modified maximal selection principle of Equation equation 14, as $N \to \infty$:*

$$\|\hat{\alpha}_{N,\theta} - \alpha_{true}\|_{\mathcal{B}} \leq \hat{C} \prod_{i=1}^{N} \rho_i \cdot \|r_0\|_{\mathcal{B}},$$

*where $\hat{C} > 0$ is a constant and $r_0$ is the initial residual.*

*Proof.* See Appendix D. □

## 4.2 TRAINING

Overall, our AFDONet-inv model is trained end-to-end by minimizing the following loss function:

$$\mathcal{L}(\theta) = \underbrace{\left|\left|\hat{\alpha}_{N,\theta} - \alpha_{\text{true}}\right|\right|_{\mathfrak{p}'}^{\mathfrak{p}'}}_{\text{reconstruction loss in } L^{\mathfrak{p}'}} + \underbrace{\omega_p \mathcal{D}_{\text{KL}}\left(\mathcal{N}(\mu_p, \sigma_p^2)\middle\| \mathcal{N}(0, I)\right)}_{\text{latent regularization loss in primal space}}$$

$$+ \underbrace{\omega_d \mathcal{D}_{\text{KL}}\left(\mathcal{N}(\mu_d, \sigma_d^2)\middle\| \mathcal{N}(0, I)\right)}_{\text{latent regularization loss in dual space}}. \quad (15)$$

The training dataset consists of various sets of PDE parameters and their corresponding PDE solutions. The Adam optimizer with a learning rate of $5 \times 10^{-4}$ is used to train our AFDONet-inv model. In actual implementation of the model, we use $\mathfrak{p}' = 1$ (corresponding to $L^1$ loss) and $\omega_p = \omega_d = 1 \times 10^{-3}$ in the loss function of Equation equation 15.

## 4.3 CONNECTIONS TO THE AFD THEORY

In AFDONet-inv, the encoder network first maps the input to its latent space, followed by a latent-to-RKBS network which finds the corresponding nearest RKBS using a feature map FM. During training, when minimizing the loss function of Equation equation 15 over different sets of PDE solutions $\{x_j\}_{j=1}^m$, we find that the optimal feature map $\text{FM}^*$ admits a finite representation $\text{FM}^*(\tilde{\alpha})(x) = \sum_{j=1}^m c_j K_z(x, x_j)$ with coefficients $c_j \in \mathbb{R}$ (see Appendix C for details). After that, the primal-dual propagation refines the value of latent variable in the Banach space and its dual, and produces the input to each layer of the dynamic CKN. This input is essentially the residual $r_i$ of AFD operation at each step $i$, as illustrated in Theorem 4.1. In this regard, the AFD-type decoder basically replicates the AFD operations. Formally, let $P_N(\alpha) = \sum_{i=1}^N \langle r_{i-1}, J(\mathscr{B}_i)\rangle_{\mathcal{B}, \mathcal{B}^*} \mathscr{B}_i$ be the $N$-term partial sum in the AFD decomposition. Then, the decoder approximation satisfies:

**Theorem 4.3.** *Let $\mathcal{B}$ be a uniformly smooth and uniformly convex Banach space, and let $\hat{\alpha}_{N,\theta}$ be the output of the AFD-type decoder after $N$ layers. Then, for any $\varepsilon > 0$, there exists parameters in the primal net and dual net, such that:*

$$\|\hat{\alpha}_{N,\theta} - P_N(\alpha)\|_{\mathcal{B}} \le \varepsilon + O\left(\frac{1}{\sqrt{N}}\right) \|\alpha\|_{\mathcal{B}}.$$

*Proof.* See Appendix E. □

## 5 EXPERIMENTS

In this section, we evaluate the performance of our AFDONet-inv on two commonly used benchmark inverse problems by conducting extensive ablation studies and comparing the solution accuracy and run time with state-of-the-art neural solvers, including NAO (Yu et al., 2024), NIPS (Liu & Yu, 2025), LNO (Wang & Wang, 2024), and MWT (Gupta et al., 2021). Each solver is trained for 1000 epochs for both benchmark problems. All experiments are performed on a B760M GAMING WIFI PLUS desktop equipped with an Intel Core i5-14600KF CPU and an NVIDIA GeForce RTX 4090 GPU (with 48GB GDDR6 memory).

## 5.1 PROBLEM SETTINGS AND DATASETS

**2-D Darcy flow.** The first inverse problem we consider is the 2-D Darcy flow problem introduced by Li et al. (2020b) and Yu et al. (2024). It takes the following form:

$$\nabla \cdot \left(a(x)\nabla u(x)\right) = f(x), \quad x \in [0, 1]^2,$$
$$u(x) = 0, \quad\quad\quad\quad\quad x \in \partial[0, 1]^2, \quad (16)$$

where $a(x)$ denotes the permeability field, and $f(x)$ is the source term. Given the solution $u(x)$ and the source term $f(x)$, here we aim to reconstruct the permeability field $a(x)$.

The details of the datasets, setups, and evaluation procedures are included in Appendix F.

**Nonlinear magnetic Schrödinger equation.** The second benchmark inverse problem involves solving the magnetic Schrödinger equation on a complex manifold $\mathcal{M}$:

$$
\begin{aligned}
\left(\Delta_A + q(|u(z)|^2)\right) u(z) = 0, & \quad z \in \mathcal{M}, \\
u(z) = f, & \quad z \in \partial\mathcal{M},
\end{aligned}
\tag{17}
$$

where $\mathcal{M} = \{z = (z_1, z_2) \in \mathbb{C}^2 : |z_1|^2 + |z_2|^2 \leq 1\}$ with boundary $\partial\mathcal{M} = S^3$, $\Delta_A = (d + iA)^*(d + iA)$ is the magnetic Laplacian (with $d$ the exterior derivative and $*$ the Hodge star with respect to the Kähler metric), $q$ is a nonlinear function, and $f$ is the boundary term. In this problem, we aim to recover the potentials $A$ and $q$ from boundary conditions in the Dirichlet-to-Neumann (DN) map $\Lambda_{A,q} : f \mapsto \partial_\nu u|_{\partial\mathcal{M}}$, where $\nu$ is the outward normal vector, and $u$ is the solution of Equation equation 17. Note that this inverse problem is more challenging to solve than the one directly given solution $u$, as the DN map only retains partial information of $u$.

## 5.2 ABLATION STUDIES

We conduct the following set of ablation studies to illustrate the need for different components in AFDONet-inv. In Scneario 1, we consider the AFDONet-inv architecture without primal-dual propagation. In Scenario 2, we investigate the impact of considering the dual space by removing the dual branch in the encoder network and the duality map $J$. Finally, in the third study, we remove both primal-dual propagation and the dual space. Results in Tables 1 and 2 indicate that incorporating primal-dual propagation and dual space is necessary for improving the overall accuracy of AFDONet-inv in terms of reducing MAE and relative $L^2$ error. To explain this, we observe that, without the dual branch, the primal-dual propagation only gives $z_{p,i}$, which is the approximation of the residual $r_i$ in the AFD theory according to Theorem 4.1. In this case, AFDONet-inv essentially approximates $\sum_{i=1}^N \langle r_i, \mathscr{B}_i \rangle \mathscr{B}_i$, which asymptotically converges under the pairing $\langle \cdot, \cdot \rangle$ with an error $O(\frac{1}{\sqrt{N}})$. Meanwhile, when we remove primal-dual propagation, the input to the AFD-type decoder is simply $\text{FM}(\tilde{\alpha})$. This way, AFDONet-inv essentially performs the operation $\sum_{i=1}^N \langle \text{FM}(\tilde{\alpha}), J(\mathscr{B}_i) \rangle_{\mathcal{B}, \mathcal{B}^*} \mathscr{B}_i$ in the Banach space, which converges with an error $O(\frac{1}{N^{k'}})$ if the PDE parameters lie in $C^{k'}$. Finally, when both primal-dual propagation and the dual branch are removed, AFDONet-inv performs the operation $\sum_{i=1}^N \langle \text{FM}(\tilde{\alpha}), \mathscr{B}_i \rangle \mathscr{B}_i$, which may not even converge since the kernels $\{\mathscr{B}_i\}_i$ are not necessarily orthogonal to each other.

Specifically, for the 2-D Darcy flow problem, the permeability field $a(x)$ on a regular domain $[0,1]^2$ is typically considered as a $L^\infty$ (or $C^{s'}$ for $s' < 0.5$ (Teng et al., 2024)) function from a statistical or computational perspective due to its irregularity. Thus, in Scenario 1, AFDONet-inv converges with an error $O(\frac{1}{N^{s'}})$ while it converges with an error $O(\frac{1}{N^{0.5}})$ in Scenario 2. This is consistent with the results shown in Table 1, in which the removal of primal-dual propagation has more significant impact on solution accuracy compared to the removal of the dual branch in the encoder. When both primal-dual propagation and dual branch are eliminated from the AFDONet-inv framework, we observe the highest MAE and relative $L^2$ error values.

For the magnetic Schrödinger equation problem on complex manifolds, the deterministic results of Krupchyk et al. (2024) indicate that both $A$ and $q$ can be relaxed to $C^\infty$. This implies a super-algebraic convergence behavior for AFDONet-inv when primal-dual propagation is removed, which explains why both MAE and relative $L^2$ error values are slightly smaller for Scenario 1 compared to Scenario 2. Last but not least, removing both primal-dual propagation and dual branch leads to the highest MAE and relative $L^2$ error due to the worst convergence behavior (or even divergence) for the resulting AFDONet-inv framework.

## 5.3 COMPARISON WITH BENCHMARK SOLVERS

Tables 3 shows the MAE, relative $L^2$ error, and computational efficiency of AFDONet-inv compared to other benchmark solvers for the 2-D Darcy flow problem. In our experiments, the architecture size

| Models | MAE | Relative $L^2$ error |
|---|---|---|
| Full | 1.82E-01 $\pm$ 6.43E-02 | 6.64E-02 $\pm$ 1.38E-03 |
| w/o prop. | 3.18E-01 $\pm$ 1.06E-01 | 7.05E-01 $\pm$ 2.19E-02 |
| w/o dual | 2.39E-01 $\pm$ 5.32E-02 | 1.07E-01 $\pm$ 4.45E-02 |
| w/o p.d. | 3.56E-01 $\pm$ 7.01E-02 | 1.93E-01 $\pm$ 3.74E-02 |

Table 1: Comparison of MAE and relative $L^2$ error in permeability field $a(x)$ on Darcy flow equation. Here and hereinafter, "Full" stands for the full AFDONet-inv model, "w/o prop." means "without primal-dual propagation" or Scenario 1 of the ablation studies, "w/o dual" means "without dual branch" or Scenario 2 of the ablation studies, and "w/o p.d." means "without both primal-dual propagation and dual branch".

| Models | MAE | Relative $L^2$ error |
|---|---|---|
| Full | 1.54E-02 $\pm$ 2.78E-03 | 1.50E-05 $\pm$ 6.23E-07 |
| w/o prop. | 7.39E-02 $\pm$ 1.81E-02 | 3.06E-04 $\pm$ 1.75E-04 |
| w/o dual | 7.83E-02 $\pm$ 2.20E-05 | 3.47E-04 $\pm$ 1.10E-05 |
| w/o p.d. | 8.01E-02 $\pm$ 6.75E-03 | 3.58E-04 $\pm$ 3.51E-05 |

Table 2: Comparison of MAE and relative $L^2$ error in potentials $A$ and $q$ on magnetic Schrödinger equation.

and training conditions are the same for all methods, and the number of parameters are different (due to the different structures present in different methods) but are in the same order of magnitude. We observe that AFDONet-inv is the second best performing solver in terms of MAE and outperforms all benchmark solvers in terms of relative $L^2$ error. Since the Darcy flow equation typically lies in $L^\infty$ space, which is a Banach space, our AFDONet-inv incorporating RKBS into our model outperforms other models. Furthermore, we also realize that for this problem, Hilbert space suffices because the permeability field $a(x)$ is often modeled with smoother priors, where the sparsity from Banach space may not be the most significant. This explains the reason that models such as NIPS also performs reasonably well.

| Models | MAE | Relative $L^2$ error | Training time |
|---|---|---|---|
| Ours | 1.82E-01 $\pm$ 6.43E-02 | 6.64E-02 $\pm$ 1.38E-03 | 2.69 |
| NAO | 1.11 $\pm$ 2.10E-01 | 7.71E-02 $\pm$ 2.09E-03 | 3.40 |
| NIPS | 1.05E-01 $\pm$ 4.71E-02 | 1.56E-01 $\pm$ 1.03E-01 | 0.96 |
| LNO | 2.78E-01 $\pm$ 3.07E-02 | 1.00 $\pm$ 3.48E-05 | 2.92 |
| MWT | 45.95$\pm$ 2.48 | 9.73E-01 $\pm$ 6.72E-02 | 1.65 |

Table 3: Comparison of MAE, relative $L^2$ error, training time (seconds per epoch) among different models on Darcy flow equation.

Meanwhile, for the nonlinear magnetic Schrödinger equation problem, we see from Table 4 that AFDONet-inv achieves remarkable performance, as it has up to one order of magnitude lower MAE and two to four orders of magnitude lower relative $L^2$ error compared to other solvers. Since the magnetic Schrödinger equation is highly nonlinear, its inverse problem is ill-posed, and non-smooth regularization such as $L^1$ penalty terms can greatly help promote sparsity when reconstructing potentials $A$ and $q$. Note that sparsity is naturally represented in an $L^1$ (Banach) space, not an $L^2$ (Hilbert) space. In this regard, our proposed AFDONet-inv solver, grounded in a novel Banach space representer theorem (Parhi & Nowak, 2021), can better capture irregular, sparse features on complex manifolds when solving ill-posed inverse problems.

Finally, in terms of computational efficiency, results in Tables 3 suggest that AFDONet-inv is competitive among all state-of-the-art benchmark solvers.

From the nonlinear magnetic Schrödinger equation results in Table 4, we can quantify the limitation of Hilbert-space assumptions. This problem is highly ill-posed, and its solution on a complex manifold benefits from sparsity-promoting regularization, which naturally matches a Banach space setting. From Table 4, it is clear that existing state-of-the-art models including NIPS, NAO, and LNO, which are implicitly or explicitly grounded in Hilbert-space frameworks, perform poorly. In

| Models | MAE | Relative $L^2$ error | Training time |
|--------|-----|----------------------|---------------|
| Ours | 1.54E-02 $\pm$ 2.78E-03 | 1.50E-05 $\pm$ 6.23E-07 | 0.52 |
| NAO | 4.65E-01 $\pm$ 5.32E-02 | 6.29E-01 $\pm$ 1.96E-01 | 2.54 |
| NIPS | 2.19E-01 $\pm$ 9.73E-02 | 1.32E-01 $\pm$ 8.40E-02 | 0.30 |
| LNO | 1.86E-01 $\pm$ 6.15E-02 | 2.89E-03 $\pm$ 3.71E-04 | 0.90 |
| MWT | 3.18E-01 $\pm$ 1.56E-01 | 7.05E-01 $\pm$ 2.46E-02 | 1.86 |

Table 4: Comparison of MAE, relative $L^2$ error, and training time (seconds per epoch) among different models on magnetic Schrödinger equation.

contrast, our AFDONet-inv, designed for Banach spaces, achieves a relative error that is two to four orders of magnitude lower than these benchmarks. This significant performance gap indicates how the Hilbert-space assumption in existing models limits their performance in practice.

## 6  CONCLUSION

In this work, we introduce AFDONet-inv to solve inverse problems in a Banach space. To the best of our knowledge, AFDONet-inv is the first neural inverse PDE solver whose design rigorously follows an established mathematical theory in every step. Each component of the neural architecture, including primal and dual nets, latent-to-RKBS network, and dynamic CKN, has a corresponding component in the AFD operation. This way, AFDONet-inv is mathematically explainable and grounded in the AFD theory and possesses several theoretical justifications and performance guarantees. Thanks to the tailored design, our AFDONet-inv solver outperforms several state-of-the-art neural operator solvers by a significant margin in terms of accuracy on inverse problems with complex manifolds and parameter space. Overall, the performance of AFDONet-inv sheds light on the systematic design of explainable neural operator frameworks.

## 7  REPRODUCIBILITY STATEMENT

All code and datasets have been either made publicly available in an anonymous repository or as a part of supplementary material to facilitate replication and verification. The experimental setup, including training steps, model configurations, and hardware details, is described in detail in the paper. We have also provided a full description of implementation details, to assist others in reproducing our experiments. Additionally, Darcy dataset, are publicly available, ensuring consistent and reproducible evaluation results.

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

# A   AFD IN RKBS

The AFD operations in RKBS are conceptually illustrated in Algorithm 1. Note that here, we do not yet consider the bias term $\gamma_i$.

---

**Algorithm 1** AFD in RKBS

---

**Require:** Target function $\alpha_{\text{true}}$, dictionary $\mathcal{D}$, duality map $J$, parameter $\rho_i$, tolerance $\varepsilon$, maximum levels of decomposition $N$

1: Initialize: $\alpha_0 \leftarrow 0$, $r_0 \leftarrow \alpha_{\text{true}}$
2: **for** $i = 1$ **to** $N$ **do**
3:    **Kernel selection:**
4:    Choose $\mathscr{B}_i \in \mathcal{D}$ such that

$$|\langle r_{i-1}, \mathscr{B}_i \rangle_{\mathcal{B},\mathcal{B}^*}| \geq \rho_i \sup_{\mathscr{B} \in \mathcal{D}} |\langle r_{i-1}, \mathscr{B} \rangle_{\mathcal{B},\mathcal{B}^*}|$$

5:    **Coefficient selection:**
6:    Compute $a_i \leftarrow \langle r_{i-1}, J(\mathscr{B}_i) \rangle_{\mathcal{B},\mathcal{B}^*}$
7:    **Update:**
8:    $\alpha_i \leftarrow \alpha_{n-1} + a_i \mathscr{B}_i$
9:    $r_i \leftarrow \alpha_{\text{true}} - \alpha_i$
10:    **if** $\|r_i\| < \varepsilon$ **then**
11:       **break**
12:    **end if**
13: **end for**

---

Here, we assume the following assumption holds. Especially, Assumption 3 is the property $(\Gamma)$ from Ganichev & Kalton (2009).

1. $\mathcal{B}$ is a uniformly smooth and uniformly convex Banach space of functions on a domain $\Omega$, with dual space $\mathcal{B}^*$.

2. $\mathcal{D}$ is defined as $\{\mathscr{B}_{a_i} | a_i \in \mathcal{B}\}$, where $\mathscr{B}_{a_i}(\cdot) = \frac{K_z(\cdot, a_i)}{\|K_z(\cdot, a_i)\|_{\mathcal{B}}}$ with poles $a_i$. The linear span of $\mathcal{D}$ is dense in $\mathcal{B}$.

3. There exists $\tilde{C} > 0$ such that for all $x \in \mathcal{B}$ with $\|x\|_{\mathcal{B}} = 1$ and $y \in \mathcal{B}$ with $\langle y, J(x) \rangle_{\mathcal{B},\mathcal{B}^*} = 0$, we have $\langle y, J(x+y) \rangle_{\mathcal{B},\mathcal{B}^*} \leq \tilde{C}(\|x+y\|_{\mathcal{B}} - 1)$.

With these assumptions, we first show that the residual $r_i$ is decreasing with respect to $i$:

**Theorem A.1.** *At each iteration $i \geq 1$, it holds that:*
$$\|r_i\|_{\mathcal{B}} < \|r_{i-1}\|_{\mathcal{B}},$$
*unless* $\langle J(r_{i-1}), \mathscr{B}_i \rangle_{\mathcal{B},\mathcal{B}^*} = 0$.

*Proof.* Define $\tilde{\phi}(a) = \|r_{i-1} - a\mathscr{B}_i\|_{\mathcal{B}}$. From Assumption 1, the uniform convexity of $\mathcal{B}$ ensures $\tilde{\phi}$ is strictly convex with a unique minimizer. Moreover, $\tilde{\phi}$ is Gâteaux differentiable in terms of its norm due to the uniform smoothness of $\mathcal{B}$. Its directional derivative is $\frac{d}{dt}\|r_{i-1} + th\|_{\mathcal{B}}\big|_{t=0} = \frac{\Re\langle J(r_{i-1}), h \rangle_{\mathcal{B},\mathcal{B}^*}}{\|r_{i-1}\|_{\mathcal{B}}}$. Now, we aim to find a direction $h$ based on $\mathscr{B}_i$ that decreases the norm, i.e., $\Re\langle J(r_{i-1}), h \rangle_{\mathcal{B},\mathcal{B}^*} < 0$. From , we choose $h = -\frac{\langle J(r_{i-1}), \mathscr{B}_i \rangle_{\mathcal{B},\mathcal{B}^*}}{|\langle J(r_{i-1}), \mathscr{B}_i \rangle_{\mathcal{B},\mathcal{B}^*}|}$, then

$$
\begin{aligned}
\Re\langle J(r_{i-1}), h \rangle_{\mathcal{B},\mathcal{B}^*} &= \Re\langle J(r_{i-1}), -\frac{\langle J(r_{i-1}), \mathscr{B}_i \rangle_{\mathcal{B},\mathcal{B}^*}}{|\langle J(r_{i-1}), \mathscr{B}_i \rangle_{\mathcal{B},\mathcal{B}^*}|}\mathscr{B}_i \rangle_{\mathcal{B},\mathcal{B}^*} \\
&= -\Re\left(\frac{\langle J(r_{i-1}), \mathscr{B}_i \rangle_{\mathcal{B},\mathcal{B}^*}}{|\langle J(r_{i-1}), \mathscr{B}_i \rangle_{\mathcal{B},\mathcal{B}^*}|}\langle J(r_{i-1}), \mathscr{B}_i \rangle_{\mathcal{B},\mathcal{B}^*}\right) \\
&= -|\langle J(r_{i-1}), \mathscr{B}_i \rangle_{\mathcal{B},\mathcal{B}^*}|.
\end{aligned}
\tag{18}
$$

If $\langle J(r_{i-1}), \mathscr{B}_i \rangle_{\mathcal{B},\mathcal{B}^*} \neq 0$, the results of Equation equation 18 is negative. So, $\tilde{\phi}(t) < \tilde{\phi}(0)$ for $t > 0$. Strict convexity then implies $\tilde{\phi}(a_i) < \tilde{\phi}(0)$ if $a_i \neq 0$, hence $\|r_i\|_{\mathcal{B}} < \|r_{i-1}\|_{\mathcal{B}}$. □

Then, the convergence of AFD in RKBS can be shown as:

**Theorem A.2.** *The sequence $\{\alpha_i\}_{i=1}^N$ generated by Algorithm 1 satisfies:*

$$\lim_{i \to \infty} \|\alpha_{true} - \alpha_i\|_{\mathcal{B}} = 0.$$

*Proof.* From Theorem A.1, the sequence $\{\|r_i\|_{\mathcal{B}}\}$ is nonincreasing and its lower bound is 0. Thus, the sequence $\{\|r_i\|_{\mathcal{B}}\}$ converges. Assume its limit $l > 0$ for contradiction. Uniform convexity from Assumption 1 implies the bounded $\{r_i\}$ has weak limit points $r'$ with $\|r'\|_{\mathcal{B}} \leq l$ by weak lower semicontinuity. By properties of the duality map $J$, we have:

$$\sup_{\mathcal{B} \in \mathcal{D}} |\langle r_{i-1}, J(\mathcal{B}) \rangle_{\mathcal{B}, \mathcal{B}^*}| = \|r_{i-1}\|_{\mathcal{B}}. \tag{19}$$

Combining Equation equation 19 and $|\langle r_{i-1}, \mathcal{B}_i \rangle_{\mathcal{B}, \mathcal{B}^*}| \geq \rho_i \sup_{\mathcal{B} \in \mathcal{D}} |\langle r_{i-1}, \mathcal{B} \rangle_{\mathcal{B}, \mathcal{B}^*}|$ leads to:

$$|\langle r_{i-1}, J(\mathcal{B}_i) \rangle_{\mathcal{B}, \mathcal{B}^*}| \geq \rho_i \|r_{i-1}\|_{\mathcal{B}}. \tag{20}$$

From Xu & Roach (1991), we have the following inequality by setting $p = 2$:

$$\|x + y\|_{\mathcal{B}}^2 \leq \|x\|_{\mathcal{B}}^2 + 2\langle J(x), y \rangle_{\mathcal{B}, \mathcal{B}^*} + 2\|x\|_{\mathcal{B}}^2 \rho \left( \frac{\|y\|_{\mathcal{B}}}{\|x\|_{\mathcal{B}}} \right), \tag{21}$$

where $\rho$ is the modulus of smoothness. By setting $x = r_{i-1}$ and $y = -a_i \mathcal{B}_i$ and assuming that $a_i$ is chosen such that $\langle J(r_{i-1}), a_i \mathcal{B}_i \rangle_{\mathcal{B}, \mathcal{B}^*} > 0$, the term $2\langle J(x), y \rangle_{\mathcal{B}, \mathcal{B}^*} = -2\langle J(r_{i-1}), a_i \mathcal{B}_i \rangle_{\mathcal{B}, \mathcal{B}^*} = -2|\langle J(r_{i-1}), a_i \mathcal{B}_i \rangle_{\mathcal{B}, \mathcal{B}^*}|$. Then, the Equation equation 21 becomes:

$$\begin{aligned} \|r_i\|_{\mathcal{B}}^2 \leq &\|r_{i-1}\|_{\mathcal{B}}^2 - 2|\langle J(r_{i-1}), a_i \mathcal{B}_i \rangle_{\mathcal{B}, \mathcal{B}^*}| \\ &+ 2\rho \left( \frac{|a_i|}{\|r_{i-1}\|_{\mathcal{B}}} \right) \|r_{i-1}\|_{\mathcal{B}}^2. \end{aligned} \tag{22}$$

Assume, for contradiction, that $\sup_{\mathcal{B} \in \mathcal{D}} |\langle r_i, J(\mathcal{B}) \rangle_{\mathcal{B}, \mathcal{B}^*}| \nrightarrow 0$. Then there exists $\delta > 0$ such that for infinitely many $i$, $\sup_{\mathcal{B} \in \mathcal{D}} |\langle r_i, J(\mathcal{B}) \rangle_{\mathcal{B}, \mathcal{B}^*}| \geq \delta$. For such $i$, the Equation equation 20 yields $|\langle r_{i-1}, J(\mathcal{B}_i) \rangle_{\mathcal{B}, \mathcal{B}^*}| \geq \rho_i \delta$. Then, we normalize $u = \frac{r_{i-1}}{\|r_{i-1}\|_{\mathcal{B}}}$ and $b_i = \frac{a_i}{\|r_{i-1}\|_{\mathcal{B}}}$, so $\frac{r_i}{\|r_{i-1}\|_{\mathcal{B}}} = u - b_i \mathcal{B}_i$. Then, the normalized version of Equation equation 20 is $|\langle u, J(\mathcal{B}_i) \rangle_{\mathcal{B}, \mathcal{B}^*}| \geq \frac{\rho_i \delta}{\|r_{i-1}\|_{\mathcal{B}}} \geq \frac{\rho_i \delta}{l}$. Next, we utilize the decomposition technique as $y = -b_i \mathcal{B}_i = \alpha J(u) + z$, where $\alpha \in \mathbb{C}$ and $\langle z, J(u) \rangle_{\mathcal{B}, \mathcal{B}^*} = 0$. Since $\langle y, J(u) \rangle = \alpha$, we have $|\langle u, J(y) \rangle_{\mathcal{B}, \mathcal{B}^*}| = |\alpha|$. And $|\alpha| \geq \rho_i \delta' b_i$ with $\delta' \geq \delta/l$. The optimality condition is $\langle J(u + y), y \rangle_{\mathcal{B}, \mathcal{B}^*} = 0$, leading to $\alpha \langle J(u + y), J(u) \rangle_{\mathcal{B}, \mathcal{B}^*} = -\langle J(u + y), z \rangle_{\mathcal{B}, \mathcal{B}^*}$. If $b_i < k\rho_i$ for a small $k > 0$, then $y$ small implies that for $\varepsilon > 0$, $\|J(u + y) - J(u)\|_{\mathcal{B}} < \varepsilon$. Therefore, we also have $\|\alpha - \langle J(u + y), z \rangle_{\mathcal{B}, \mathcal{B}^*}\|_{\mathcal{B}} < \varepsilon$. From Assumption 3, since $\langle z, J(u) \rangle_{\mathcal{B}, \mathcal{B}^*} = 0$, $|\langle z, J(u+y) \rangle_{\mathcal{B}, \mathcal{B}^*}| \leq \tilde{C}(\|u + y\|_{\mathcal{B}} - 1)$. With $b_i \to 0$, by smoothness, we obtain $\|u + y\|_{\mathcal{B}} - 1 = O(b_i)$, so $|\alpha| \to 0$, contradicting $|\alpha| \geq \rho_i \delta' b_i$. Thus, it holds that $b_i \geq k\rho_i$ for $k > 0$ depending on $\tilde{C}$ and the smoothness modulus. Substitute them into Equation equation 21 leads to the term $-2|\langle J(r_{i-1}), a_i \mathcal{B}_i \rangle_{\mathcal{B}, \mathcal{B}^*}|$ is at least $c'\rho_i^2 \|r_{i-1}\|_{\mathcal{B}}^2$ for the constant $c'$, and the term $2\|r_{i-1}\|_{\mathcal{B}}^2 \rho(b_i)$ is $o(\rho_i^2 \|r_{i-1}\|_{\mathcal{B}}^2)$ since $\rho(\cdot) = o(\cdot)$, yielding

$$\|r_i\|_{\mathcal{B}}^2 \leq \|r_{i-1}\|_{\mathcal{B}}^2 - c\rho_i^2 \|r_{i-1}\|_{\mathcal{B}}^2, \tag{23}$$

for $c > 0$. Assume that there exists $\delta > 0$ such that for infinitely many indices $i_k$ (with $\|r_{i_k}\|_{\mathcal{B}}^2 \to l^2$), the decrease satisfies

$$\|r_{i_k}\|_{\mathcal{B}}^2 - \|r_{i_k+1}\|_{\mathcal{B}}^2 \geq c\rho_{i_k}^2 \|r_{i_k}\|_{\mathcal{B}}^2 \geq c\rho^2 l^2/2 =: d > 0, \tag{24}$$

where the last inequality holds for sufficiently large $k$ since $\|r_{i_k}\|_{\mathcal{B}}^2 \to l^2$ and $\rho_{i_k} \geq \rho > 0$. Let $S = \sum_{k=1}^{\infty} (\|r_{i_k}\|_{\mathcal{B}}^2 - \|r_{i_k+1}\|_{\mathcal{B}}^2) \geq \sum_{k=1}^{\infty} d = \infty$. Since the overall sequence decreases by at most $\|r_1\|_{\mathcal{B}}^2 - l^2 < \infty$, but includes $S = \infty$, which leads to a contradiction. Therefore, for large $M$, the remaining decrease after $i_M$ is at least $\sum_{k>M} d = \infty$, but it is upper bounded by $\|r_{i_M}\|_{\mathcal{B}}^2 - l^2 < \infty$. Between $i_k$ and $i_{k+1}$, the decrease is nonnegative. Thus, after $m$ such steps,

$$\|r_{i_{k+m}}\|_{\mathcal{B}}^2 \leq \|r_{i_k}\|_{\mathcal{B}}^2 - md \to -\infty \quad (m \to \infty), \tag{25}$$

which is impossible for norms. The exponential form follows recursively. If decrease $\geq c\rho^2 \|r_i\|_{\mathcal{B}}^2$ at each of $m$ steps, finally it leads to

$$\|r_{i+m}\|_{\mathcal{B}}^2 \leq (1 - c\rho^2)^m \|r_i\|_{\mathcal{B}}^2 \to 0, \tag{26}$$

contradicting convergence to $l^2 > 0$.

Thus, $\sup_{\mathscr{B} \in \mathcal{D}} |\langle r_i, J(\mathscr{B}) \rangle_{\mathcal{B}, \mathcal{B}^*}| \to 0$. For weak limit $r'$ of $\{r_{i_k}\}$, $\langle r_{i_k}, J(\mathscr{B}) \rangle \to \langle r', J(\mathscr{B}) \rangle = 0$ for all $\mathscr{B} \in \mathcal{D}$. Bijectivity of $J$ and density of $\operatorname{span} \mathcal{D}$ imply $\{J(\mathscr{B})\}$ generates dense functionals, forcing $r' = 0$. All weak limits are 0, but $\|r_i\|_{\mathcal{B}} \to l > 0 = \|0\|_{\mathcal{B}}$, contradicting weak lower semicontinuity unless $l = 0$. Hence, $\lim_{i \to \infty} \|r_i\|_{\mathcal{B}} = 0$, so $\lim_{i \to \infty} \|\alpha_{\text{true}} - \alpha_i\|_{\mathcal{B}} = 0$. $\qquad\square$

## B    Proof of Theorem 4.1

Without loss of generality, we consider the case $\mathcal{B} = L^p([0, L]^d)$ for $1 < p < \infty$. Before giving the complete proof of Theorem 4.1, we introduce some lemmas as follows.

**Lemma B.1.** *Let $\mathcal{B} = L^p([0, L]^d)$ for $1 < p < \infty$, a uniformly smooth Banach space admitting a Fourier transform $\mathcal{F}$. Let $(\psi_\xi)_{\xi \in \mathbb{Z}^d}$ be the normalized Fourier basis $\psi_\xi(x) = e^{2\pi i \xi \cdot x / L}$, with Fourier coefficients $\hat{f}(\xi) = \mathcal{F}(f)(\xi)$. Define the partial sum operators $S_n : \mathcal{B} \to \mathcal{B}$ by $S_n f = \sum_{|\xi|_\infty \leq n} \hat{f}(\xi) \psi_\xi$, where $|\xi|_\infty = \max_j |\xi_j|$. Then, there exists $C_p' > 0$ (depending on $p$ and $d$, but independent of $n$) such that $\|S_n f\|_{\mathcal{B}} \leq (C_p')^d \|f\|_{\mathcal{B}}$ for all $f \in \mathcal{B}$ and all $n \in \mathbb{N}$.*

*Proof.* First, we prove this lemma for $d = 1$ (torus $\mathbb{T} \cong [0, L]$) and then extend it to higher dimensions. For $d = 1$, we have:

$$
\begin{aligned}
S_n f(x) &= \sum_{|k| \leq n} \hat{f}(k) e^{2\pi i k x / L} \\
&= f * D_n \\
&:= f * \left( \sum_{|k| \leq n} e^{2\pi i k x / L} \right).
\end{aligned}
\tag{27}
$$

Define $\tilde{f}(x) = -i \sum_{k \in \mathbb{Z}} \operatorname{sgn}(k) \hat{f}(k) e^{2\pi i k x / L}$, $\|\tilde{f}\|_{\mathcal{B}} \leq C_p \|f\|_{\mathcal{B}}$ indicates $\sup_n \|S_n\|_{\mathcal{B}} < \infty$ holds. Consider the Riesz projection $P_+(f) = \sum_{k \geq 0} \hat{f}(k) e^{2\pi i k x / L}$. Then, $P_+(f) = \frac{1}{2}(f + i\tilde{f}) + \frac{1}{2}\hat{f}(0)$, so boundedness of $P_+$ on $L^p$ is equivalent to the boundedness of $\tilde{\cdot}$. Since

$$
\begin{aligned}
S_n f &= \hat{f}(0) + \sum_{k=1}^n \left( \hat{f}(k) e^{2\pi i k x / L} + \hat{f}(-k) e^{-2\pi i k x / L} \right) \\
&= 2 \operatorname{Re}(P_+^{(n)}) - \hat{f}(0),
\end{aligned}
\tag{28}
$$

where $P_+^{(n)}(f) = \sum_{k=0}^n \hat{f}(k) e^{ikx}$ and

$$
\|P_+^{(n)}(f) - f\|_{\mathcal{B}} \leq (M + 1)\varepsilon
\tag{29}
$$

for a constant $M > 0$ and any $\varepsilon > 0$ from Miao (2014). From the generalized M. Riesz Theorem Berkson & Gillespie (1985), it follows $\|\tilde{f}\|_{\mathcal{B}} \leq C_p \|f\|_{\mathcal{B}}$, which by the identity $P_+(f) = \frac{1}{2}(f + i\tilde{f}) + \frac{1}{2}\hat{f}(0)$, implies that the Riesz projection $P_+$ is also bounded on $\mathcal{B}$, with

$$
\|P_+ f\|_{\mathcal{B}} \leq C_p' \|f\|_{\mathcal{B}}.
\tag{30}
$$

Since $P_+^{(n)} f \to P_+ f$ in $\mathcal{B}$ norm as $n \to \infty$, it follows that $\{P_+^{(n)}\}_{n \in \mathbb{N}}$ is uniformly bounded on $\mathcal{B}$, i.e.,

$$
\sup_n \|P_+^{(n)}\|_{\mathcal{B}} < \infty.
\tag{31}
$$

Therefore, one can obtain

$$
\begin{aligned}
\|S_n f\|_{\mathcal{B}} &\leq 2\|P_+^{(n)} f\|_{\mathcal{B}} + |\hat{f}(0)| \cdot \|1\|_{\mathcal{B}} \\
&\leq 2C_p \|f\|_{\mathcal{B}} + C\|f\|_{\mathcal{B}} = C_p' \|f\|_{\mathcal{B}}
\end{aligned}
\tag{32}
$$

for a constant $C > 0$ using the triangle inequality.

For $d$-dimensional case, we define the $d$-dimensional kernel $D_n^d(\mathbf{x}) = \prod_{j=1}^d D_n(x_j)$ and the $d$-dimensional operator $S_n^d = S_n \otimes \cdots \otimes S_n$. From Equation equation 32, we have:

$$\|S_n^d f\|_{\mathcal{B}} \le (C_p')^d \|f\|_{\mathcal{B}}. \tag{33}$$

$\square$

**Lemma B.2.** *Let $\mathcal{P}$ denote the space of trigonometric polynomials, i.e., finite linear combinations of plane waves $\psi_\xi(x) = e^{2\pi i \xi \cdot x / L}$ for $\xi \in \mathbb{Z}^d$. Then, $\mathcal{P}$ is dense in $\mathcal{B}$, meaning for any $f \in \mathcal{B}$ and $\epsilon > 0$, there exists $p \in \mathcal{P}$ such that $\|f - p\|_{\mathcal{B}} < \epsilon$. For any $g \in \mathcal{P}$ of degree at most $n$, it holds that $S_n g = g$, where $S_n g = \sum_{|\xi| \le n} \hat{g}(\xi) \psi_\xi$.*

*Proof.* Since $\mathbb{T}^d$ is compact with finite measure, continuous functions $C(\mathbb{T}^d)$ are dense in $\mathcal{B}(\mathbb{T}^d)$ for any $1 \le p < \infty$. For $f \in \mathcal{B}$, by Lusin's theorem (every measurable function is nearly continuous), for any $\epsilon > 0$, there exists a compact $K \subset \mathbb{T}^d$ with $\mu(\mathbb{T}^d \setminus K) < \epsilon$ and $f|_K$ continuous, hence can be approximated by continuous $g$ with $\|f - g\|_{\mathcal{B}} < \epsilon$.

Next, we prove that $\mathcal{P}$ is dense in $C(\mathbb{T}^d)$. The Stone-Weierstrass theorem states that if $\mathcal{A}$ is a subalgebra of $C(X)$ that separates points (for any distinct $x, y \in X$, there exists $f \in \mathcal{A}$ with $f(x) \ne f(y)$) and contains constants, then $\mathcal{A}$ is dense in $C(X)$ under the sup-norm. Following this, considering the algebra $\mathcal{A} = \mathcal{P}_{\mathbb{R}}$ of real parts of $\mathcal{P}$, $\mathcal{A}$ contains constants and separates points: for distinct $x, y \in \mathbb{T}^d$, choose $\xi \in \mathbb{Z}^d$ such that $\xi \cdot (x - y) \not\equiv 0 \pmod 1$, then $\cos(2\pi \xi \cdot x / L) \ne \cos(2\pi \xi \cdot y / L)$. Moreover, $\mathcal{A}$ is closed under multiplication. Thus, $\mathcal{A}$ is dense in $C(\mathbb{T}^d; \mathbb{R})$. Next, for $C(\mathbb{T}^d; \mathbb{C})$, density follows by approximating real and imaginary parts separately, since $\mathcal{P}$ includes both cosines and sines. Therefore, given $\mathcal{P}$ dense in $C(\mathbb{T}^d)$ and $C(\mathbb{T}^d)$ dense in $\mathcal{B}$, implies $\mathcal{P}$ dense in $\mathcal{B}$ from transitivity of density.

Then, for ( $g \in \mathcal{P}$ with degree at most $n$, we have $\hat{g}(\xi) = 0$ for all $|\xi| > n$. Therefore, it follows:

$$S_n g = \sum_{|\xi| \le n} \hat{g}(\xi) \psi_\xi = \sum_{\xi \in \mathbb{Z}^d} \hat{g}(\xi) \psi_\xi = g, \tag{34}$$

the last equality follows

$$g = \sum_{\xi \in \mathbb{Z}^d} \langle g, \psi_\xi \rangle \psi_\xi, \tag{35}$$

where $\langle g, \psi_\xi \rangle = \hat{g}(\xi)$. $\square$

**Lemma B.3.** *With the Assumption 1, by assuming that $\mathcal{B}$ admits a Fourier transform $\mathcal{F}$ that is well-defined and invertible on a periodic domain $[0, L]^d$), for the Fourier basis of plane waves $\psi_\xi(x) = e^{2\pi i \xi \cdot x / L}$ denoted as $(\psi_\xi)_{\xi \in \mathbb{Z}^d}$, and the biorthogonal dual functionals $(\varphi_\xi)_{\xi \in \mathbb{Z}^d} \subset \mathcal{B}^*$ satisfying $\hat{f}(\xi) = \mathcal{F}(f)(\xi) = \langle f, \varphi_\xi \rangle_{\mathcal{B}, \mathcal{B}^*}$ for any $f \in \mathcal{B}$, then the inverse Fourier transform satisfies $\mathcal{F}^{-1}(\hat{f}(\xi)) = \sum_{\xi \in \mathbb{Z}^d} \hat{f}(\xi) \psi_\xi(x)$ in the sense that the series converges to $f$ in the $\mathcal{B}$-norm, i.e., $\lim_{n \to \infty} \left\| f - \sum_{|\xi| \le n} \hat{f}(\xi) \psi_\xi \right\|_{\mathcal{B}} = 0$.*

*Proof.* Following Lin et al. (2022), one can show that the plane waves $\psi_\xi(x) = e^{2\pi i \xi \cdot x / L}$ span a dense subspace of trigonometric polynomials in $\mathcal{B}$. The corresponding dual functionals $\varphi_\xi \in \mathcal{B}^*$ extract the Fourier coefficients via the pairing $\hat{f}(\xi) = \langle f, \varphi_\xi \rangle_{\mathcal{B}, \mathcal{B}^*}$. Biorthogonality, given by $\langle \psi_\xi, \varphi_\eta \rangle_{\mathcal{B}, \mathcal{B}^*} = \delta_{\xi\eta}$, is ensured by the RKBS structure and follows from the spectral theorem applied to the associated kernel integral operator. Then, we define the operator $S_n : \mathcal{B} \to \mathcal{B}$ via $S_n f = \sum_{|\xi| \le n} \hat{f}(\xi) \psi_\xi$. From Lemma B.1, $\|S_n\|_{\mathcal{B} \to \mathcal{B}}$ is bounded by a constant $(C_p')^d$ independent of $n$.

From Lemma B.2 and uniform boundedness principle, $S_n f \to f$ in $\mathcal{B}$-norm for all $f \in \mathcal{B}$. $\square$

**Remark B.1.** *An important consequence is the Hausdorff–Young inequality:*

$$\left( \sum_\xi |\hat{f}(\xi)|^q \right)^{1/q} \le C \|f\|_{\mathcal{B}}, \quad \text{where } \frac{1}{p} + \frac{1}{q} = 1, \tag{36}$$

*which provides a bound on the Fourier coefficients in $\ell^q$, which facilitates convergence of the Fourier series. For finite modes $\Lambda \subset \mathbb{Z}^d$, the sum is finite and exact, and the infinite case follows by taking limits as $|\Lambda| \to \infty$. Thus, the reconstruction defines the inverse Fourier transform $\mathcal{F}^{-1}(\hat{f}(\xi)) = \sum_\xi \hat{f}(\xi) \psi_\xi(x)$ in $\mathcal{B}$.*

**Lemma B.4.** *It holds that*

$$\mathcal{S}(f)(x) = \sum_{\xi \in \Lambda} w_\xi \langle f, \varphi_\xi \rangle_{\mathcal{B}, \mathcal{B}^*} \psi_\xi(x),$$

*where $w_\xi = W(\xi)$, $\varphi_\xi \in \mathcal{B}^*$ are dual bases, and $\psi_\xi \in \mathcal{B}$ are biorthogonal primal bases with $\langle \psi_\xi, \varphi_\eta \rangle_{\mathcal{B}, \mathcal{B}^*} = \delta_{\xi\eta}$.*

*Proof.* In spectral methods for RKBS, one can show that the Fourier transform diagonalizes convolution operators Kovachki et al. (2021). Specifically, we assume a biorthogonal Fourier basis $(\psi_\xi, \varphi_\xi)_{\xi \in \mathbb{Z}^d}$, where $\psi_\xi(x) = e^{2\pi i \xi \cdot x / L}$ for domain $[0, L]^d$, and $\varphi_\xi$ are dual functionals satisfying the biorthogonality from the reproducing property: $\langle f, \varphi_\xi \rangle = \hat{f}(\xi) = \mathcal{F}(f)(\xi)$. Then, the spectral convolution applies pointwise multiplication in frequency space:

$$\widehat{\mathcal{S}(f)}(\xi) = \chi(\xi) W(\xi) \hat{f}(\xi). \tag{37}$$

Taking the inverse Fourier transform on Equation equation 37 leads to:

$$\mathcal{S}(f)(x) = \sum_{\xi \in \Lambda} W(\xi) \hat{f}(\xi) \psi_\xi(x), \tag{38}$$

where $\chi(\xi) = 1$ for $\xi \in \Lambda$ and 0 otherwise. Substituting $\hat{f}(\xi) = \langle f, \varphi_\xi \rangle_{\mathcal{B}, \mathcal{B}^*}$ into Equation equation 38 leads to

$$\mathcal{S}(f)(x) = \sum_{\xi \in \Lambda} W(\xi) \langle f, \varphi_\xi \rangle_{\mathcal{B}, \mathcal{B}^*} \psi_\xi(x). \tag{39}$$

Biorthogonality $\langle \psi_\xi, \varphi_\eta \rangle = \delta_{\xi\eta}$ follows from the Fourier basis orthogonality in the dual pairing, ensured by the RKBS structure Li et al. (2022). $\square$

**Remark B.2.** *Equation equation 39 is nonlocal because the kernel involves global frequency modes rather than localized supports.*

**Lemma B.5.** *For any continuous operator $T : \mathcal{B} \to \mathcal{B}$ and $\delta > 0$, there exists a set of parameters of $W(\xi)$ in Equation equation 9 such that $\|\mathcal{S}(g) - T(g)\|_{\mathcal{B}} < \delta$ for all $g$ in a bounded subset of $\mathcal{B}$.*

*Proof.* First, we define integral operators $T(f)(x) = \int K_T(x, y) f(y) dy$, with kernel $K_T$ continuous. Then, fixing a bounded set $G \subset \mathcal{B}$, we say $G$ is weakly compact since $\mathcal{B}$ is uniformly smooth and reflexive. By Arzelà-Ascoli theorem, continuous kernels $K_T(x, y)$ and thus operator $T(f)$ can be approximated uniformly on $G$. We denote the Fourier operator $T_M(f) = \mathcal{F}^{-1}[\sum_{|\xi| \le M} \hat{K}_T(\xi) \mathcal{F}(f)(\xi)]$, where $\hat{K}_T(\xi)$ is the Fourier transform of the kernel $K_T(x, y)$. From universal approximation theorem Kovachki et al. (2021), we have:

$$\|T - T_M\|_{\mathcal{B}} \to 0, \text{ as } M \to \infty \tag{40}$$

Next, by setting $\Lambda = \{\xi : |\xi| \le M\}$, $\chi(\xi) = 1_\Lambda(\xi)$, and $W(\xi) = \hat{K}_T(\xi)$, it follows

$$\|\mathcal{S} - T_M\|_{\mathcal{B}} \to 0, \text{ as } M \to \infty. \tag{41}$$

Finally, substituting Equation equation 40 to Equation equation 41 leads to:

$$\begin{aligned}
\|\mathcal{S}(g) - T(g)\|_{\mathcal{B}} &\le \|\mathcal{S}(g) - T_M(g) + T_M(g) - T(g)\|_{\mathcal{B}} \\
&\le \|\mathcal{S}(g) - T_M(g)\|_{\mathcal{B}} + \|T(g) - T_M(g)\|_{\mathcal{B}} \\
&< \frac{\delta}{2} + \frac{\delta}{2} = \delta,
\end{aligned} \tag{42}$$

which completes the proof. $\square$

**Lemma B.6.** *Define $\mathcal{C}(f)(x) = \int_\Omega \delta(x - y) \cdot W \cdot f(y) \, dy$, where $\delta$ is the Dirac delta distribution. The integral is well-defined in the distributional sense.*

*Proof.* First, we consider the single channel ($C = 1$, $W = 1$). In this case, we have $\mathcal{C}(f)(x) = f(x) = \int_\Omega \delta(x - y)f(y)\,dy$, where the integral is the pairing $\langle \delta_x, f \rangle_{\mathcal{B},\mathcal{B}^*}$ in the dual space since $\delta_x \in \mathcal{B}^*$ for spaces admitting point evaluations. Next, for multi-channel $f = (f^{(1)}, \ldots, f^{(C)})$, we extend $\mathcal{C}(f)^{(c)}(x)$ to $\sum_{c'=1}^{C} W_{cc'} f^{(c')}(x)$. Then, it follows

$$
\begin{aligned}
\mathcal{C}(f)^{(c)}(x) &= \sum_{c'=1}^{C} W_{cc'} f^{(c')}(x) \\
&= \sum_{c'=1}^{C} W_{cc'} \int_\Omega \delta(x - y)f^{(c')}(y)\,dy \\
&= \int_\Omega \delta(x - y)(Wf(y))^{(c)}\,dy,
\end{aligned}
\tag{43}
$$

where $Wf(y)$ applies the matrix pointwise. Equation equation 43 becomes to

$$
\mathcal{C}(f)(x) = \int_\Omega \delta(x - y) \cdot W \cdot f(y)\,dy,
\tag{44}
$$

for $f(y) \in \mathbb{R}^C$. $\qquad\square$

**Remark B.3.** *$\mathcal{C}(f)(x)$ is a local projection because the kernel $\delta(x - y)W$ has support only at $y = x$ which projects onto the span of channels without spatial smearing. In practice, for a discrete domain, it reduces to matrix multiplication at pixels. For a continuous domain, it is actually the distributional convolution with a point mass.*

**Lemma B.7.** *Finite-rank operators are dense in the space of compact operators.*

*Proof.* It is equivalent to show for any $\delta > 0$, there exists a finite-rank operator $K_m$ such that $\|T - K_m\|_\mathcal{B} < \delta$. Assume the operator $T$ is compact, then its image on a unit ball $B$ has compact closure. Therefore, for any $x \in \mathcal{B}_B$, there exists $v_i \in \mathcal{B}$, such that $\|T(x) - v_i\|_\mathcal{B} < \delta$ for any $\delta > 0$. Now, we define a projection $\pi_V : \text{range}(T) \to V$, which maps each $T(x)$ to its best approximation in $V$. It is well-defined because $V$ is finite-dimensional and $\mathcal{B}$ is a Banach space. Then, one can also define $K_m := \pi_V \circ T$, which is a linear and bounded operator. And the range of $K_m$ lies in the span of $v_i$, implying that $K_m$ is finite-rank. This way, one can verify $\|T(f) - K_m(f)\|_\mathcal{B} = \|T(f) - \pi_V(T(f))\|_\mathcal{B} \leq \delta$ for any $\delta > 0$ and $f \in \mathcal{B}$. Since $K_m(f) \in \text{span}\{v_1, \ldots, v_m\}$, one can formulate it as $K_m(f) = \sum_{i=1}^{m} a_i(f)\,v_i$, for linear functionals $a_i$. $\qquad\square$

**Lemma B.8.** *$\mathcal{C}$ is finite-rank. For any operator $T : \mathcal{B} \to \mathcal{B}$ following $T = I - P$, where $P$ is a projection, and $\delta > 0$, there exists $W_{cc'}$ such that $\|\mathcal{C}(g) - T(g)\|_\mathcal{B} < \delta$ for all $g$ in a bounded subset $G \subset \mathcal{B}$, with the error decaying as the matrix rank (channel dimension) $C$ increases.*

*Proof.* First, we show that $\mathcal{C}$ is finite-rank. Since $\mathcal{B}$ is an RKBS, the evaluation functional is continuous, for $f \in \mathcal{B}$, $f(x) = \langle f, K(\cdot, x) \rangle_{\mathcal{B},\mathcal{B}^*}$ holds, and $K(\cdot, x) \in \mathcal{B}^*$ satisfies $\|K(\cdot, x)\|_{\mathcal{B}^*} < \infty$. Let $\mathcal{B}^C = \mathcal{B} \otimes \mathbb{R}^C$, $f = (f^{(1)}, \ldots, f^{(C)})$ has evaluations $f(x) = (f^{(1)}(x), \ldots, f^{(C)}(x)) \in \mathbb{R}^C$, with

$$
f^{(c)}(x) = \langle f^{(c)}, K(\cdot, x) \rangle_{\mathcal{B},\mathcal{B}^*}.
\tag{45}
$$

For each channel $c = 1, \ldots, C$, we have

$$
\mathcal{C}(f)^{(c)}(x) = \sum_{c'=1}^{C} W_{cc'} f^{(c')}(x).
\tag{46}
$$

The reproducing property follows

$$
f^{(c')}(x) = \langle f^{(c')}, K(\cdot, x) \rangle_{\mathcal{B},\mathcal{B}^*} = \langle f^{(c')}, \delta_x \rangle_{\mathcal{B},\mathcal{B}^*}
\tag{47}
$$

in the distributional sense as long as $\mathcal{B}$ embeds into a space where $\delta_x$ is defined. Then, it holds that

$$
\langle f^{(c')}, \delta_x \rangle_{\mathcal{B},\mathcal{B}^*} = \int_\Omega \delta(x - y)f^{(c')}(y)\,dy
\tag{48}
$$

in the weak sense. Therefore, we have:

$$
\begin{aligned}
\mathcal{C}(f)^{(c)}(x) &= \sum_{c'=1}^{C} W_{cc'} \langle f^{(c')}, \delta_x \rangle_{\mathcal{B},\mathcal{B}^*} \\
&= \sum_{c'=1}^{C} W_{cc'} \int_{\Omega} \delta(x-y) f^{(c')}(y) \, dy.
\end{aligned}
\tag{49}
$$

Next, Equation equation 49 becomes:

$$
\mathcal{C}(f) = \sum_{c=1}^{C} \sum_{c'=1}^{C} W_{cc'} \langle f, \mathscr{B}_{c'} \rangle_{\mathcal{B}^C} \mathscr{B}_c,
\tag{50}
$$

where $\langle f, \mathscr{B}_{c'} \rangle_{\mathcal{B}^C} = \langle f^{(c')}, \mathscr{B}_c \rangle_{\mathcal{B},\mathcal{B}^*}$ for the basis $\mathscr{B}_c$ of $\mathcal{B}$. The rank of $\mathcal{C}(f)$ is at most $c \times c' \leq C^2$, so $\mathcal{C}(f)$ is finite-rank. We note that if an operator is finite-rank, it is bounded and compact in Banach space $\mathcal{B}$.

Moreover, we consider the operator $T = I - P$, where $P$ is a projection operator. Assume that $P$ is finite-rank and thus compact, and then the operator $T$ is also compact. Since $T$ is compact, for any $\delta > 0$, there exists finite-rank approximation $K_m = \sum_{i=1}^{m} \sigma_i \langle \cdot, u_i \rangle_{\mathcal{B}^C} v_i$, where $u_i, v_i \in \mathcal{B}^C$, such that

$$
\|T(g) - K_m(g)\|_{\mathcal{B}} < \frac{\delta}{\sup_{g \in G} \|g\|_{\mathcal{B}^C}}
\tag{51}
$$

for $g \in G$ from Lemma B.7.

Then, by denoting $A = I - P$, from Equation equation 51, we have:

$$
\begin{aligned}
\|\mathcal{C}(g) - T(g)\|_{\mathcal{B}^C} &= \left\| \int_{\Omega} \delta(x-y)(W - A(y)) g(y) \, dy \right\|_{\mathcal{B}^C} \\
&= \left( \int_{\Omega} \|(W - A(y)) g(y)\|_{\mathbb{R}^C}^2 \, dy \right)^{1/2} \\
&\leq \sup_{y} \|W - A(y)\|_{\mathcal{B}^C} \left( \int_{\Omega} \|g(y)\|_{\mathbb{R}^C}^2 \, dy \right)^{1/2} \\
&\leq \epsilon \|g\|_{\mathcal{B}^C} < \delta,
\end{aligned}
\tag{52}
$$

by choosing $\epsilon < \delta/(2 \sup \|g\|_{\mathcal{B}^C})$. $\qquad \square$

Now, we give the full proof of Theorem 4.1 assuming GELU is globally Lipschitz with constant $L_{\text{GELU}}$.

*Proof.* We start with the residual:

$$
\begin{aligned}
r_i &= \alpha_{\text{true}} - \alpha_i = (\alpha_{\text{true}} - \alpha_{i-1}) - (\alpha_i - \alpha_{i-1}) \\
&= r_{i-1} - \langle r_{i-1}, J(\mathscr{B}_i) \rangle_{\mathcal{B},\mathcal{B}^*} \mathscr{B}_i.
\end{aligned}
\tag{53}
$$

Then, let $T(g) = \langle g, J(\mathscr{B}_i) \rangle_{\mathcal{B},\mathcal{B}^*} \mathscr{B}_i$, from Lemma B.5, one can obtain:

$$
\|\mathcal{S}(z_{p,i-1}) - \langle r_{i-1}, J(\mathscr{B}_i) \rangle_{\mathcal{B},\mathcal{B}^*} \mathscr{B}_i\|_{\mathcal{B}} < \frac{\varepsilon}{4 L_{\text{GELU}}}.
\tag{54}
$$

Next, from Lemma B.8, we can also get:

$$
\begin{aligned}
&\|\mathcal{C}(z_{p,i-1}) - (r_{i-1} - z_{p,i-1}) - \langle r_{i-1}, J(\mathscr{B}_i) \rangle_{\mathcal{B},\mathcal{B}^*} \mathscr{B}_i\|_{\mathcal{B}} \\
&\qquad\qquad\qquad\qquad < \frac{\varepsilon}{4 L_{\text{GELU}}}.
\end{aligned}
\tag{55}
$$

By adjusting the parameters in GELU (i.e., $\gamma, \beta$), one can show:

$$
\begin{aligned}
&\|\text{BN}(z_{p,i-1} + \mathcal{S}(z_{p,i-1}) + \mathcal{C}(z_{p,i-1})) - (r_{i-1} \\
&\quad - \langle r_{i-1}, J(\mathscr{B}_i) \rangle_{\mathcal{B},\mathcal{B}^*} \mathscr{B}_i)\|_{\mathcal{B}} < \frac{\varepsilon}{2 L_{\text{GELU}}}.
\end{aligned}
\tag{56}
$$

By Lipschitz continuity of GELU, we have:

$$\|z_{p,i} - r_i\|_{\mathcal{B}} < L_{\text{GELU}} \cdot \left( \frac{\varepsilon}{4L_{\text{GELU}}} + \frac{\varepsilon}{4L_{\text{GELU}}} + \frac{\varepsilon}{2L_{\text{GELU}}} \right) < \varepsilon. \tag{57}$$

Since $J$ is continuous, one can show $\|z_{d,i} - J(r_i)\|_{\mathcal{B}}$ at the same way. $\qquad\square$

## C  THE OPTIMAL FEATURE MAP FM$^*$

We denote the training data points here as $(x_i, y_i)$. The empirical risk minimization in the RKBS $\mathcal{B}$ is utilized in the form $\min_{f \in \mathcal{B}} \sum_{j=1}^m \ell(\text{FM}(x_j), y_j) + \lambda\|\text{FM}\|_{\mathcal{B}}$, where FM refers to the possible feature maps, $\ell$ is the loss function which is assumed to be convex, continuous, and Lipschitz, and $\lambda \geq 0$ is the regularization parameter. Since $\mathcal{B}$ is reflexive, we know that the optimal feature map FM$^*$ as a minimizer exists. Then, we calculate it by solving $\partial(\sum \ell(\text{FM}(x_j), y_j)) + \lambda\partial\|\text{FM}\|_{\mathcal{B}}) = 0$ following:

$$\begin{aligned}
\partial\left(\sum \ell(\text{FM}(x_j), y_j)\right) &= \sum_j \partial_u \ell(u, y_j)|_{u=\text{FM}(x_j)} \cdot \partial\text{FM}(x_j) \\
&= \sum_j \partial_u \ell(u, y_j)|_{u=\text{FM}(x_j)} \cdot K_z^*(x_j, \cdot),
\end{aligned} \tag{58}$$

where $\partial\text{FM}(x_j) = K_z^*(x_j, \cdot) \in \mathcal{B}^*$ from evaluation functional. Thus, $\partial\ell(\text{FM}) \subset \text{span}\{K_z^*(x_j, \cdot)\}_{j=1}^m$ in dual.

On the other hand, we have $\partial\|\text{FM}\|_{\mathcal{B}} = \frac{J(\text{FM})}{\|\text{FM}\|_{\mathcal{B}}}$ if FM $\neq 0$. We admit FM$^*$ follows the form $\sum_{j=1}^m c_j K_z(x, x_j)$, where $c_j$ needs to be determined. Substituting it into the gradient condition leads to:

$$\sum_k c_k K_z(x_k, x_j) + \lambda\langle J(\text{FM}^*), K_z(\cdot, x_j)\rangle_{\mathcal{B},\mathcal{B}^*} = -\partial\ell(\text{FM}^*(x_j), y_j), \tag{59}$$

which is a linear system can be solved accordingly.

## D  PROOF OF THEOREM 4.2

Before providing the full proof of Theorem 4.2, we give several definitions and lemmas first.

**Definition D.1.** *A Banach space $\mathcal{B}$ is uniformly convex if there exists a function $\delta : [0, 2] \to [0, 1]$, the modulus of convexity, such that for all $u, v \in \mathcal{B}$ with $\|u\|_{\mathcal{B}} = \|v\|_{\mathcal{B}} = 1$ and $\|u - v\|_{\mathcal{B}} \geq \tau$,*

$$\left\|\frac{u+v}{2}\right\|_{\mathcal{B}} \leq 1 - \delta(\tau), \quad \delta(\tau) > 0 \text{ for } \tau > 0.$$

*$\mathcal{B}$ is uniformly smooth if the modulus of smoothness $\rho(\tau) = \sup\{\frac{\|u+\tau v\|_{\mathcal{B}} + \|u-\tau v\|_{\mathcal{B}}}{2} - 1 : \|u\|_{\mathcal{B}} = \|v\|_{\mathcal{B}} = 1\}$ satisfies $\rho(\tau) = o(\tau)$ as $\tau \to 0$.*

**Lemma D.1.** *For any $\eta > 0$, there exist parameters in the primal-dual propagation such that $\|z_{p,i} - r_i\|_{\mathcal{B}} < \eta$ and $\|z_{d,i} - J(r_i)\|_{\mathcal{B}^*} < \eta$ for each $i$. Therefore, it holds that*

$$|\langle z_{p,i}, J(\mathcal{B}_{i+1})\rangle_{\mathcal{B},\mathcal{B}^*} - \langle r_i, J(\mathcal{B}_{i+1})\rangle_{\mathcal{B},\mathcal{B}^*}| < \eta.$$

*Proof.* This lemma follows directly from Theorem 4.1. $\qquad\square$

**Lemma D.2.** *In the AFD operations, the residuals satisfy*

$$\|r_i\|_{\mathcal{B}}^2 \leq \rho_i^2 \|r_{i-1}\|_{\mathcal{B}}^2,$$

*where $\rho_i^2 = 1 - \delta\left(2\frac{|\langle r_{i-1}, J(\mathcal{B}_i)\rangle_{\mathcal{B},\mathcal{B}^*}|}{\|r_{i-1}\|_{\mathcal{B}}}\right) < 1$, and $\mathcal{B}_i$ is chosen greedily as $\max_{\mathcal{B}_j \in \mathcal{D}} |\langle r_{i-1}, J(\mathcal{B}_j)\rangle_{\mathcal{B},\mathcal{B}^*}|$.*

*Proof.* For $u = \frac{r_{i-1}}{\|r_{i-1}\|_{\mathcal{B}}}$ and $v = \frac{\langle r_{i-1}, J(\mathscr{B}_i) \rangle_{\mathcal{B},\mathcal{B}^*} \mathscr{B}_i}{\|r_{i-1}\|_{\mathcal{B}}}$, it follows

$$\left\| \frac{r_i}{\|r_{i-1}\|_{\mathcal{B}}} \right\|_{\mathcal{B}} = \left\| u - \frac{\langle u, J(\mathscr{B}_i) \rangle_{\mathcal{B},\mathcal{B}^*} \mathscr{B}_i}{\|r_{i-1}\|_{\mathcal{B}}} \right\|_{\mathcal{B}} = \|u - v\|_{\mathcal{B}}. \tag{60}$$

Since $\|u - v\|_{\mathcal{B}}^2 \leq 1 - \delta\left(2\|u\|_{\mathcal{B}}\|v\|_{\mathcal{B}}\right)$, we have:

$$\|u - v\|_{\mathcal{B}}^2 \leq 1 - \delta\left(2\|v\|_{\mathcal{B}}\right)$$
$$\leq 1 - 2\delta\left( \frac{|\langle r_{i-1}, J(\mathscr{B}_{a_i}) \rangle_{\mathcal{B},\mathcal{B}^*}|}{\|r_{i-1}\|_{\mathcal{B}}} \right) := 1 - \rho_i. \tag{61}$$

Equation equation 61 becomes to:

$$\|r_i\|_{\mathcal{B}}^2 \leq \rho_i^2 \|r_{i-1}\|_{\mathcal{B}}^2. \tag{62}$$

Iterating Equation equation 62 leads to:

$$\|r_N\|_{\mathcal{B}} \leq \left( \prod_{i=1}^{N} \rho_i \right) \|r_0\|_{\mathcal{B}}. \tag{63}$$

$\square$

**Lemma D.3.** *The bias terms satisfy*

$$\sum_{i=1}^{N} |\gamma_i| \leq \left( 1 - \prod_{i=1}^{N} \rho_i \right) \|r_0\|_{\mathcal{B}} + N\varepsilon,$$

*where the bias term satisfying* $\gamma_i \leq |\langle z_{p,i-1}, J(\mathscr{B}_i) \rangle| - \rho_i \sup_{\mathscr{B}_j \in \mathcal{D}} |\langle z_{p,i-1}, J(\mathscr{B}_j) \rangle|$, *and* $\|z_{p,i-1} - r_{i-1}\|_{\mathcal{B}} < \varepsilon$.

*Proof.* Assume $\mathcal{D}$ is rich in $\mathcal{B}$, we have:

$$|\gamma_i| = |\langle z_{p,i-1}, J(\mathscr{B}_i) \rangle| - \rho_i \sup_{\mathscr{B}_j} |\langle z_{p,i-1}, J(\mathscr{B}_j) \rangle|$$
$$\leq (1 - \rho_i)\|z_{p,i-1}\|_{\mathcal{B}}. \tag{64}$$

Now, from Theorem 4.1, one can have:

$$\|z_{p,i-1}\|_{\mathcal{B}} \leq \|r_{i-1}\|_{\mathcal{B}} + \|z_{p,i-1} - r_{i-1}\|_{\mathcal{B}} \leq \|r_{i-1}\|_{\mathcal{B}} + \varepsilon \tag{65}$$

for $\varepsilon > 0$.

From Lemma D.2, Equation equation 65 becomes to:

$$\|z_{p,i-1}\|_{\mathcal{B}} \leq \prod_{j=1}^{i-1} \rho_j \|r_0\|_{\mathcal{B}} + \varepsilon. \tag{66}$$

Substituting Equation equation 64 into Equation equation 66 leads to:

$$|\gamma_i| \leq (1 - \rho_i) \left( \prod_{j=1}^{i-1} \rho_j \|r_0\|_{\mathcal{B}} + \varepsilon \right). \tag{67}$$

Then, we sum both sides of Equation equation 64 over $i = 1$ to $N$ and get:

$$\sum_{i=1}^{N} |\gamma_i| \leq \sum_{i=1}^{N} (1 - \rho_i) \left( \prod_{j=1}^{i-1} \rho_j \|r_0\|_{\mathcal{B}} + \varepsilon \right)$$
$$= \|r_0\|_{\mathcal{B}} \sum_{i=1}^{N} (1 - \rho_i) \prod_{j=1}^{i-1} \rho_j + \varepsilon \sum_{i=1}^{N} (1 - \rho_i) \tag{68}$$
$$\leq \|r_0\|_{\mathcal{B}} \sum_{i=1}^{N} (1 - \rho_i) \prod_{j=1}^{i-1} \rho_j + \varepsilon N.$$

For the first term in Equation equation 68, it follows

$$\sum_{i=1}^{N}(1-\rho_i)\prod_{j=1}^{i-1}\rho_j = \sum_{i=1}^{N}\left(\prod_{j=1}^{i-1}\rho_j - \prod_{j=1}^{i}\rho_j\right)$$
$$= (1-\rho_1) + (\rho_1 - \rho_1\rho_2) + (\rho_1\rho_2 - \rho_1\rho_2\rho_3)$$
$$+ \cdots + (\prod_{j=1}^{N-1}\rho_j - \prod_{j=1}^{N}\rho_j) \quad (69)$$
$$= 1 - \prod_{j=1}^{N}\rho_j.$$

Putting everything together, we arrive:

$$\sum_{i=1}^{N}|\gamma_i| \le \left(1 - \prod_{i=1}^{N}\rho_i\right)\|r_0\|_{\mathcal{B}} + N\varepsilon, \quad (70)$$

which completes the proof. $\square$

Now, we are safe to give the full proof of Theorem 4.2.

*Proof.* To start with, we consider

$$\|\hat{\alpha}_{N,\theta} - \alpha_{\text{true}}\|_{\mathcal{B}} = \left\|\sum_{i=1}^{N}\langle z_{p,i}, J(\mathcal{B}_i)\rangle_{\mathcal{B},\mathcal{B}^*}\mathcal{B}_i + \sum_{i=1}^{N}\gamma_i - \right.$$
$$\left. \sum_{i=1}^{\infty}\langle r_{i-1}, J(\mathcal{B}_i)\rangle_{\mathcal{B},\mathcal{B}^*}\mathcal{B}_i \right\|_{\mathcal{B}}$$
$$\le \sum_{i=1}^{N}|\langle z_{p,i} - r_{i-1}, J(\mathcal{B}_i)\rangle_{\mathcal{B},\mathcal{B}^*}| \cdot \|\mathcal{B}_i\|_{\mathcal{B}} \quad (71)$$
$$+ \sum_{i=1}^{N}|\gamma_i| + \left\|\sum_{i=N+1}^{\infty}\langle r_{i-1}, J(\mathcal{B}_i)\rangle_{\mathcal{B},\mathcal{B}^*}\mathcal{B}_i\right\|_{\mathcal{B}}.$$

From Lemma D.1, one can have:

$$|\langle z_{p,i} - r_{i-1}, J(\mathcal{B}_i)\rangle| < \eta, \quad (72)$$

for any $\eta > 0$, which implies

$$\sum_{i=1}^{N}|\langle z_{p,i} - r_{i-1}, J(\mathcal{B}_i)\rangle_{\mathcal{B},\mathcal{B}^*}| < N\eta = \frac{\varepsilon}{2} \quad (73)$$

by setting $\eta = \frac{\varepsilon}{2N}$. By Theorem A.2, we have:

$$\alpha_{\text{true}} = \sum_{i=1}^{\infty}\langle r_{i-1}, J(\mathcal{B}_i)\rangle_{\mathcal{B},\mathcal{B}^*}\mathcal{B}_i, \quad (74)$$

so the tail is

$$\left\|\sum_{i=N+1}^{\infty}\langle r_{i-1}, J(\mathcal{B}_i)\rangle_{\mathcal{B},\mathcal{B}^*}\mathcal{B}_i\right\|_{\mathcal{B}} = \|r_N\|_{\mathcal{B}} \le \prod_{i=1}^{N}\rho_i\|r_0\|_{\mathcal{B}} \quad (75)$$

from Lemma D.2. Next, from Lemma D.3, the bias term $\gamma_i$ satisfying:

$$\sum_{i=1}^{N}|\gamma_i| \le \left(1 - \prod_{i=1}^{N}\rho_i\right)\|r_0\|_{\mathcal{B}} + N\eta$$
$$= \left(1 - \prod_{i=1}^{N}\rho_i\right)\|r_0\|_{\mathcal{B}} + \frac{\varepsilon}{2}. \quad (76)$$

By choosing $\gamma_i \le (1 - \rho_i) \prod_{j=1}^{i-1} \rho_j \|r_0\|_{\mathcal{B}}$, we have:

$$\|\hat{\alpha}_{N,\theta} - \alpha_{\text{true}}\|_{\mathcal{B}} \le \frac{\varepsilon}{2} + \prod_{i=1}^{N} \rho_i \|r_0\|_{\mathcal{B}} + \frac{\varepsilon}{2}$$

$$= \varepsilon + \prod_{i=1}^{N} \rho_i \|r_0\|_{\mathcal{B}}, \tag{77}$$

which completes the proof. □

## E    PROOF OF THEOREM 4.3

Before proving Theorem 4.3, we give a lemma first.

**Lemma E.1.** *Let $\ell$ be a convex and $L$-Lipschitz loss function on a compact convex subset $\mathcal{B}_0 \subseteq \mathcal{B}$, and let $\Phi$ be a $\mu$-strongly convex mirror map on $\mathcal{B}_0$. Then, the mirror descent algorithm (MDA) with projected updates achieves a convergence rate of $O(1/\sqrt{N})$ for $N$ iterations:*

$$\ell(f_N) - \ell(f^*) \le O\left(\frac{1}{\sqrt{N}}\right),$$

*where $f^*$ is the minimizer, and the constant depends on $L$, $\mu$, and the diameter of $\mathcal{B}_0$.*

*Proof.* It follows from Kumar et al. (2024) by setting $g_N = g_{N-1} - \eta \partial_{f_{N-1}} \mathcal{L}$ and $f_N = \Pi_{\mathcal{B}_0}^{\Phi}((\partial \Phi)^{-1}(g_N))$, where $\Pi$ is the Bregman projection. □

**Remark E.1.** *We remark that, AFD can be interpreted as a greedy variant of mirror descent in RKBS, where each layer corresponds to a descent step with duality pairing approximating subgradients, and the normalized kernels $\mathscr{B}_i$ are utilized to select the directions. In AFD, the greedy selection maximizes the projection $|\langle r_{i-1}, J(\mathscr{B}_a)\rangle|$, which is equivalent to a subgradient descent step in the dual space, with the mirror map $\Phi(f) = \frac{1}{2}\|f\|_{\mathcal{B}}^2$. Furthermore, $r_i = r_{i-1} - \langle r_{i-1}, J\rangle_{\mathcal{B},\mathcal{B}^*} \mathscr{B}_i$ is a projected mirror descent step.*

With this, now we prove Theorem 4.3.

*Proof.* Define the loss function $\ell(f) = \|\alpha - f\|_{\mathcal{B}}$ on $\mathcal{B}_0 = \{f \in \text{span}(\mathcal{D}) : \|f\|_{\mathcal{B}} \le \|\alpha\|_{\mathcal{B}}\}$. Since $\mathcal{B}$ is uniformly convex, $\mathcal{B}_0$ is compact. The function $\ell(f)$ is convex and Lipschitz:

$$|\ell(f) - \ell(g)| = |\|\alpha - f\|_{\mathcal{B}} - \|\alpha - g\|_{\mathcal{B}}| \le \|f - g\|_{\mathcal{B}}. \tag{78}$$

One can know that the minimizer is $f^* = \alpha$ and $\ell(f^*) = 0$ holds in this case. In AFD, we have

$$r_i = r_{i-1} - \langle r_{i-1}, J(\mathscr{B}_i)\rangle_{\mathcal{B},\mathcal{B}^*} \mathscr{B}_i \tag{79}$$

and

$$P_i(\alpha) = P_{i-1}(\alpha) + \langle r_{i-1}, J(\mathscr{B}_i)\rangle_{\mathcal{B},\mathcal{B}^*} \mathscr{B}_i, \tag{80}$$

where $\mathscr{B}_i$ maximizes $|\langle r_{i-1}, J(\mathscr{B}_j)\rangle_{\mathcal{B},\mathcal{B}^*}|$. This way, AFD corresponds to a greedy mirror descent step, where the subgradient direction is approximated by $J(\mathscr{B}_i)$. Denote $f_i = P_i(\alpha)$, so $r_i = \alpha - f_i$. Furthermore, Equation equation 80 is equivalent to:

$$f_i = f_{i-1} + \eta_i J^*(\partial \ell(f_{i-1})), \tag{81}$$

where $\eta_i = \langle r_{i-1}, J(\mathscr{B}_i)\rangle$, and $J^* : \mathcal{B}^* \to \mathcal{B}$ is the inverse duality map since $\mathcal{B}$ is reflexive.

From Lemma E.1, considering $\ell(f) = \|\alpha - f\|_{\mathcal{B}}$ on $\mathcal{B}_0$, we have

$$\ell(f_N) - \ell(f^*) = \|\alpha - f_N\|_{\mathcal{B}_0} \le O\left(\frac{1}{\sqrt{N}}\right), \tag{82}$$

where the diameter $\text{diam}(\mathcal{B}_0) \le 2\|\alpha\|_{\mathcal{B}}$. Since $f_N = P_N(\alpha)$, and $\ell(f^*) = 0$,

$$\|P_N(\alpha) - \alpha\|_{\mathcal{B}_0} = \|\alpha - f_N\|_{\mathcal{B}_0} \le O\left(\frac{1}{\sqrt{N}}\right). \tag{83}$$

Then, we generalize it to $\|\alpha\|_\mathcal{B}$ by scaling diameter:

$$\|P_N(\alpha) - \alpha\|_\mathcal{B} \leq O\left(\frac{1}{\sqrt{N}}\right) \|\alpha\|_\mathcal{B}. \tag{84}$$

Finally, from Theorem 4.2, we obtain:

$$
\begin{aligned}
\|\hat{\alpha}_{N,\theta} - P_N(\alpha)\|_\mathcal{B} &\leq \|\hat{\alpha}_{N,\theta} - \alpha\|_\mathcal{B} + \|\alpha - P_N(\alpha)\|_\mathcal{B} \\
&\leq \hat{C} \prod_{i=1}^{N} \rho_i \cdot \|r_0\|_\mathcal{B} + O\left(\frac{1}{\sqrt{N}}\right) \|\alpha\|_\mathcal{B} \\
&< \varepsilon + O\left(\frac{1}{\sqrt{N}}\right) \|\alpha\|_\mathcal{B},
\end{aligned}
\tag{85}
$$

which completes the proof. $\qquad\square$

## F  DETAILS OF DATASETS, SETUPS, AND EVALUATION PROCEDURES

Two datasets used in this work include Darcy flow (public dataset from Li et al. (2020b)) and nonlinear magnetic Schrödinger. For Darcy flow dataset, the coefficients $a$ are generated following a measure $\mu$ defined as $\mu = \psi\left(\mathcal{N}\left(0, (-\Delta + 9I)^{-2}\right)\right)$, where the operator $(-\Delta + 9I)^{-2}$ utilizes a Neumann boundary condition. The field $a$ is constructed to be piecewise constant with random geometry and a fixed contrast of 4, determined by the mapping $\psi(x) = 12$ for $x > 0$ and $\psi(x) = 3$ for $x \leq 0$. Solutions $u$ are generated using a second-order finite difference scheme on a high-resolution $241 \times 241$ grid.

For nonlinear magnetic Schrödinger dataset, we just need the Dirichlet-to-Neumann (DN) map without needing to generate PDE solutions. That is, the DN map data serves as the observation, rather than a solution field $u$. This data is generated by specifying the functional class of the potentials $A$ (magnetic) and $q$ (scalar) and the boundary term $f$ on the complex manifold $\mathcal{M}$. The potentials $A$ and $q$ represent the unknown parameters, and the DN map $\Lambda\_A, q : f \mapsto \partial\_\nu u|\_\partial\mathcal{M}$ is computed by numerically solving the highly nonlinear Schrödinger equation for various input boundary terms $f$ and then calculating the resulting normal derivative $\partial_\nu u$ at the boundary 2. The complete dataset consists of pairs of the unknown potentials $(A, q)$ and their corresponding simulated DN maps.

## G  ADDITIONAL EXPERIMENTS

Here, we conduct an additional experiment considering the magnetic Schrödinger equation problem on a regular rectangular domain $[0, 1] \times [0, 1]$. The results are shown in Table 6.

| Models | MAE | Relative $L^2$ error |
|--------|-----|----------------------|
| Ours | 3.20E-02 | 5.30E-05 |
| NAO | 8.09E-01 | 1.01 |
| NIPS | 2.07E-01 | 8.05E-02 |
| LNO | 4.64E-01 | 1.86E-01 |
| MWT | 2.82E-01 | 1.00 |

Table 5: Comparison of MAE and relative $L^2$ error among different models on magnetic Schrödinger equation on $[0, 1] \times [0, 1]$.

Furthermore, we also explore the effect of data augmentation. Given that the data augmentation process is achieved by random permutations. Here, we implement 100 random permutations on top of the training data containing 6000 solution samples. We compare our model to NAO and NIPS, whose performance heavily relies on data augmentation.

## H  ARCHITECTURE AND IMPLEMENTATION DETAILS

The architecture and implementations details can be found in the source code uploaded along with the submission.

| Models | MAE | Relative $L^2$ error | Training time |
|--------|-----|----------------------|---------------|
| Ours | 9.94E-03 | 1.20E-05 | 2.52 |
| NAO | 5.98E-02 | 6.11E-03 | 2.54 |
| NIPS | 4.81E-02 | 4.29E-03 | 3.26 |

Table 6: Comparison of MAE, relative $L^2$ error and training time (seconds per epoch) among different models on magnetic Schrödinger equation on $[0, 1] \times [0, 1]$ under 100 random permutations.

