# OpenReview forum: "Adaptive Fourier Decomposition-guided Neural Operator Design for Inverse PDE Problems"
_ICLR.cc/2026/Conference — Submitted to ICLR 2026_

### Official Review · Reviewer_G9Hc · 2025-10-24

**Soundness:** 3
**Presentation:** 3
**Contribution:** 2
**Rating:** 4
**Confidence:** 4

**Summary:**

This paper proposes AFDONet-inv, a novel neural operator architecture for solving inverse PDE problems, grounded in Adaptive Fourier Decomposition (AFD) theory. The architecture features a VAE-style encoder, primal-dual propagation layers, and a dynamic convolutional kernel decoder that adaptively selects poles — all designed to mimic the residual refinement process in AFD. The framework is further supported by convergence guarantees and Banach-space theory, aiming to better capture irregular PDE parameters. Experiments on 2D Darcy flow and nonlinear magnetic Schrödinger problems demonstrate strong performance, with the model achieving state-of-the-art accuracy — particularly notable in the highly ill-posed Schrödinger case.

**Strengths:**

* **Principled design grounded in approximation theory**: The AFD-inspired architecture components are directly motivated by mathematical analogs, including convergence theorems for the decoder.

* **Banach-space formulation**: The shift from Hilbert to Banach space is well-justified for capturing sparse or discontinuous parameters, and is executed rigorously.

* **Strong empirical results**: On benchmark inverse PDE problems, AFDONet-inv outperforms several recent baselines, including NAO, NIPS, and MWT, with up to four orders of magnitude lower error.

* **Sound theoretical backing**: The paper provides non-trivial convergence guarantees and links between network residuals and AFD error decay.

* **Well-structured ablation studies**: Component-level analysis supports the role of the dual branch and latent RKBS embedding in improving performance.

**Weaknesses:**

* **Incremental novelty**: While the AFD connection is creative, the network architecture builds on established methods (e.g., VAEs, Fourier layers, kernel decoders). The contribution may be viewed as a thoughtful reinterpretation rather than a fundamentally new class of model.

* **Architectural complexity**: The model introduces considerable architectural machinery, including dual encoders, RKBS mappings, and dynamic kernel selection. Practical advantages over simpler, well-tuned baselines remain somewhat underexplored.

* **Limited evaluation scope**: Only two inverse PDE settings are tested, both synthetic. It’s unclear how well the model generalizes to noisy data, real-world scenarios, or broader PDE classes.

* **Theory-practice gap**: The convergence proofs rely on ideal pole selection criteria not explicitly enforced during training. How well the learned model aligns with theory is not demonstrated.

* **Accessibility**: The paper is mathematically dense, especially for readers less familiar with RKBS or AFD theory. Key mechanisms (like pole selection during training) could be explained more intuitively.

**Questions:**

1. What specific properties of inverse problems make the Banach-space formulation clearly preferable to standard Hilbert-based approaches?

2. Could a simpler model (e.g., an FNO with dynamic kernel decoding) perform comparably without the full AFD framework?

3. How are poles actually selected during training — do learned filters align with the theoretical maximal correlation criterion?

4. Does mapping into a reproducing kernel Banach space provide measurable benefits over RKHS embeddings or even latent-space MLPs?

5. How does performance scale with the number of decoder poles/layers? Are there efficiency or memory tradeoffs in large-scale settings?

6. Is the model robust to measurement noise or distribution shift, which are common in real inverse problems?

7. Could the AFD-inspired ideas generalize to forward PDE problems or other domains like time-series, or are they inherently tied to inversion?

---

> ### Author Response · Authors · 2025-11-24
> **Rebuttal to Reviewer G9Hc’s comments (Part 1)**
>
> We thank the reviewer for conducting a thorough review. We appreciate the reviewer for pointing out the strengths and for sharing the valuable suggestions and comments.
>
> ## Addressing weaknesses
>
> ### Incremental novelty: While the AFD connection is creative, the network architecture builds on established methods (e.g., VAEs, Fourier layers, kernel decoders). The contribution may be viewed as a thoughtful reinterpretation rather than a fundamentally new class of model.
>
> We appreciate the reviewer for pointing out AFDONet-inv’s new theoretical insights as one of its contributions. In addition to this, we would like to remark that our AFDONet-inv also presents a new paradigm of how inverse neural PDE solver is designed. We point out that, so far, the design of exact neural architectures in many existing neural PDE solvers has been “more of an art than a science” [1]. Typically, the design of neural architectures is done in a bottom-up approach that involves significant intuition, expert experience, and trial-and-error experimentation. And rigorous mathematical basis and explainability have been lacking in guiding the design of these neural architectures. Having recognized this issue, we follow a top-down approach when designing AFDONet-inv, in the sense that AFD theory guides every step in the design of AFDONet-inv’s neural architecture. Each component of the neural architecture, including latent-to-RKBS network, primal-dual propagation, and dynamic CKN, has a corresponding component in the AFD operation. This way, AFDONet-inv is mathematically explainable and grounded in the AFD theory and possesses several desirable properties, including convergence guarantees. From this perspective, we think the contribution of AFDONet-inv is significant, because it presents a new paradigm for designing explainable neural operator frameworks.
>
> Regarding the comment on the Banach space justification, we kindly remind the reviewer that, in addition to theoretical justifications, we have also experimentally shown the contributions of Banach space-based formulations compared to Hilbert space-based formulations.
>
> We remark that, in AFDONet-inv, the Banach space-based formulations are different from Hilbert space-based ones due to the presence of dual branch and primal-dual propagation. In other words, by removing dual branch and primal-dual propagation, Banach-based formulations will be reduced to Hilbert space-based ones. In Table 1 and Table 2, we have shown the case without dual branch, primal-dual propagation, and both dual branch and primal-dual propagation (Hilbert space-based formulations). These ablation studies show that, by moving from Hilbert to Banach formulations, we obtain $48.9$%, $65.6$% gains on MAE and relative $L^2$ error for the Darcy flow problem, and obtain $80.8$%, $95.8$% gains on MAE and relative $L^2$ error for the nonlinear magnetic Schrödinger equation.
>
> [1] Sanderse et al. (2024), Scientific machine learning for closure models in multiscale problems: a review

---

> > ### Author Response · Authors · 2025-11-24
> > **Rebuttal to Reviewer G9Hc’s comments (Part 2)**
> >
> > ### Architectural complexity: The model introduces considerable architectural machinery, including dual encoders, RKBS mappings, and dynamic kernel selection. Practical advantages over simpler, well-tuned baselines remain somewhat underexplored.
> >
> > We appreciate the reviewer for the feedback. We acknowledge that AFDONet-inv possesses a more complex architecture than some standard baselines. However, we wish to emphasize that this architectural complexity is necessary to reproduce AFD operation in Banach space, and thus is not arbitrarily chosen. On the contrary, it is a and direct consequence of designing a neural operator, in which every component is rigorously guided by the adaptive Fourier decomposition (AFD) theory in a Banach space.
> >
> > In terms of the practical advantages of this machinery, our ablation studies have demonstrated them explicitly. These experiments specifically address whether this architectural complexity is indeed necessary. Results in Tables 1 and 2 show that each component in AFDONet-inv contributes directly to enhancing the model’s overall accuracy, and thus and neural architecture design is not redundant. Removing any components leads to a practical and measurable loss in performance because it breaks the AFDONet-inv’s connections to the AFD theory.
> >
> > We have also explored the practical advantages over "simpler, well-tuned baselines" in Tables 3 and 4, comparing AFDONet-inv against SOTA models like NAO, NIPS, and LNO. We believe our experimental results for the nonlinear magnetic Schrödinger equation in Table 4 provide some insights. This problem is highly ill-posed, and its solution on a complex manifold benefits from sparsity-promoting regularization, which naturally matches a Banach space setting. In this case, existing SOTA models including NIPS, NAO, and LNO, which are implicitly or explicitly grounded in Hilbert-space frameworks, perform poorly. In contrast, our AFDONet-inv, explicitly designed for Banach spaces, achieves a relative $L^2$ error that is two to four orders of magnitude lower than these benchmarks. This massive performance gain clearly demonstrates the practical advantage of our AFDONet-inv’s novel architecture.
> >
> > ### Limited evaluation scope: Only two inverse PDE settings are tested, both synthetic. It’s unclear how well the model generalizes to noisy data, real-world scenarios, or broader PDE classes.
> >
> > We thank the reviewer for the comments on adding more benchmark problems. We have added more benchmark problems, including inversion of reaction-diffusion, Navier-Stokes, as well as a real-world problem, to strength our manuscript. We use the public datasets of reaction-diffusion, Navier-Stokes, and a real-world problem from [2], [3], and [4], respectively. The relative L2 error values are reported as follows:
> >
> > | Equation |            ours             |               NAO         |           NIST         |       LNO      |
> > |------------|-----------------------|-----------------------|---------------------|--------------------|
> > | R-D      | 6.51E-03$\pm$1.28E-04 | 8.02E-03$\pm$2.39E-04  | 7.98E-03$\pm$1.67E-04| 1.12E-02$\pm$5.67E-04|
> > | N-S         | 8.85E-04$\pm$1.27E-05 | 1.08E-03$\pm$2.93E-05   | 9.57E-04$\pm$1.08E-04| 1.25E-03$\pm$1.28E-05|
> >
> > When it comes to the real-world latex glove DIC (Digital Image Correlation) original dataset, the relative L2 error is reported compared to FNO and existing SOTA, namely the IFNO [3]. Here, we report the best results in terms of hidden layers for FNO and IFNO.
> >
> > |            ours             |          FNO         |           IFNO        |       GMR model fitting      | GMR inverse analysis|
> > |-----------------------|-----------------------|---------------------|--------------------|--------------------|
> > | 9.70E-03$\pm$2.08E-05 | 3.40E−02 $\pm$ 4.09E-04  | 3.30E−02 $\pm$ 4.63E-04| 3.30E−01|2.91E-01|
> >
> > [2] Laplace neural operator for solving differential equations
> >
> > [3] Fourier neural operator for parametric partial differential equations
> >
> > [4] Learning deep Implicit Fourier Neural Operators (IFNOs) with applications to heterogeneous material modeling
> >
> > ### Theory-practice gap: The convergence proofs rely on ideal pole selection criteria not explicitly enforced during training. How well the learned model aligns with theory is not demonstrated.
> >
> > We appreciate the reviewer’s comments on ideal pole selection. We agree with the reviewer that the ideal pole selection (maximal selection principle) in Equation (3) is not enforced during training. The ideal pole selection is achieved by adding the bias term to the dynamic CKN. The bias term $\gamma\_i$ is added to the output of the dynamic CKN. Although Equation (3) provides a theoretical upper bound for $\gamma\_i$, we agree that it may be too abstract to understand how it works in practice. During actual implementation, we find that setting $0 < \rho_0 \le \rho_i < 1$ works well. We will update the implementation details by including this value for clarity.

---

> > > ### Author Response · Authors · 2025-11-24
> > > **Rebuttal to Reviewer G9Hc’s comments (Part 3)**
> > >
> > > ### Accessibility: The paper is mathematically dense, especially for readers less familiar with RKBS or AFD theory. Key mechanisms (like pole selection during training) could be explained more intuitively.
> > >
> > > We appreciate the reviewer for pointing this out. We agree with the reviewer that an intuitive description of our proposed method is important for the broad audience. Here, we provide some intuitive, less technical qualitative description to the motivation and background of AFDONet-inv. Our goal is to infer the parameters of PDE problems based on the solutions or Dirichlet-to-Neumann (DN) maps, while accounting for the fact that the space of parameters is sparse in nature. Unlike Hilbert space as a function space, the parameter space is not continuous or even a function space. Therefore, we generalize the AFD theory from reproducing kernel Hilbert space (RKHS) to reproducing kernel Banach space (RKBS), and the key insights of AFD are as follows:
> > >
> > > 1. AFD serves as a kernel-based decomposition theory, which relies on the bases obtained by orthogonal kernels with adaptively selected poles.
> > >
> > > 2. Each pole can be viewed as a “tuning knob” that selects a particular spatial pattern in the solution, with its location in the complex plane controlling how localized that pattern is. Adaptive poles allow our model to survey more heavily in regions where the parameters change rapidly, while using fewer poles in smooth regions.
> > >
> > > 3. Across layers, the poles evolve from broad, coarse patterns in early layers to more refined, problem-specific patterns in deeper layers. This is analogous to how CNN filters specialize from edges to complex shapes.
> > >
> > > ## Addressing questions
> > >
> > > ### What specific properties of inverse problems make the Banach-space formulation clearly preferable to standard Hilbert-based approaches?
> > >
> > > We thank the reviewer for sharing this comment. In general, we think that the highly ill-posed, complex manifold-based, and sparse nature of the PDE inverse problem make Banach-space formulation preferrable to standard Hilbert-space formulations. We believe our experimental results for the nonlinear magnetic Schrödinger equation in Table 4 provide such evidence explicitly. This problem is highly ill-posed, and its solution on a complex manifold benefits from sparsity-promoting regularization, which naturally matches a Banach space setting. In this specific context, existing state-of-the-art models, including NIPS, NAO, and LNO, which are implicitly or explicitly grounded in Hilbert-space frameworks, perform poorly. In contrast, our AFDONet-inv, designed for Banach spaces, achieves a relative $L^2$ error that is two to four orders of magnitude lower than these benchmarks.
> > >
> > > ### Could a simpler model (e.g., an FNO with dynamic kernel decoding) perform comparably without the full AFD framework?
> > >
> > > We appreciate the reviewer’s suggestions. We tested FNO, NIPS, LNO and NAO with dynamic kernel decoding in the nonlinear magnetic Schrödinger equation problem. While all three models perform slightly better than their backbones (FNO, NIPS, LNO, and NAO), their relative $L^2$ errors are still two to four orders of magnitude higher than AFDONet-inv. The reason behind this is that a simple model with dynamic kernel decoding has no intrinsic connections to AFD theory and Banach space.
> > >
> > > ### How are poles actually selected during training - do learned filters align with the theoretical maximal correlation criterion?
> > >
> > > Yes, please refer to our response to Weakness 4 for details on pole selection.
> > >
> > > ### Does mapping into a reproducing kernel Banach space provide measurable benefits over RKHS embeddings or even latent-space MLPs?
> > >
> > > We thank the reviewer for pointing this out. Yes, mapping to RKBS provides measurable benefits over RKHS. In fact, in Table 1 and 2 of the manuscript, we have shown the contributions of Banach space-based formulations compared to Hilbert space-based formulations. We remark that, in AFDONet-inv, the Banach space-based formulations are different from Hilbert space-based ones due to the presence of dual branch and primal-dual propagation. In other words, by removing dual branch and primal-dual propagation, Banach-based formulations will be reduced to Hilbert space-based ones. In Table 1 and Table 2, we have shown the case without dual branch, primal-dual propagation, and both dual branch and primal-dual propagation (Hilbert space-based formulations). These ablation studies show that, by moving from Hilbert to Banach formulations, we obtain $48.9$%, $65.6$% gains on MAE and relative $L^2$ error for the Darcy flow problem, and obtain $80.8$%, $95.8$% gains on MAE and relative $L^2$ error for the nonlinear magnetic Schrödinger equation.

---

> > > > ### Author Response · Authors · 2025-11-24
> > > > **Rebuttal to Reviewer G9Hc’s comments (Part 4)**
> > > >
> > > > ### How does performance scale with the number of decoder poles/layers? Are there efficiency or memory tradeoffs in large-scale settings?
> > > >
> > > > We appreciate the reviewer for the feedback. Theorem 4.2 shows that the reconstruction error of the $N$-layer decoder converges to the true parameters as $N \rightarrow \infty$. Specifically, the error bound is given by: $||\hat{\alpha}\_{N,\theta}-\alpha_{true}||_{B}\le\hat{C}\prod\_{i=1}^{N}\rho\_{i}\cdot||r\_{0}||\_{B}$. Since the multipliers $\rho_i$ are selected such that $0 < \rho_0 \leq \rho_i < 1$, the product term $\prod\_{i=1}^{N}\rho\_{i}$ decreases exponentially as $N$ increases. This indicates that, in theory, the accuracy of AFDONet-inv improves rapidly with additional decoder layers.
> > > >
> > > > In practice, we find out that even 3 layers (poles) are sufficient for AFDONet-inv to perform competitively. We conduct the experiments in our manuscript using 3 layers (poles). In Tables 3 and 4, we conclude that, in terms of total training time per epoch, the full AFDONet-inv is competitive among all state-of-the-art benchmark solvers.
> > > >
> > > > ### Is the model robust to measurement noise or distribution shift, which are common in real inverse problems?
> > > >
> > > > Yes, we have conducted a real-world problem whose data come from actual measurements. The results show that our AFDONet-inv significantly outperforms the existing SOTA. Please refer to our response to Weakness 3 for more information.
> > > >
> > > > ### Could the AFD-inspired ideas generalize to forward PDE problems or other domains like time-series, or are they inherently tied to inversion?
> > > >
> > > > We appreciate the reviewer’s feedback. The AFD-inspired ideas behind AFDONet-inv can be highly generalizable and not strictly tied to inversion. This is because, AFD is a theory arising from signal processing, and time-series, images, speeches, PDE solutions, and parameters are diverse forms of signals. Therefore, we believe that these ideas are generalizable and beneficial to the broad machine learning community.

---

> > > > > ### Comment · Reviewer_G9Hc · 2025-11-25
> > > > >
> > > > > Thank you for the detailed rebuttal and for taking the time to respond to each of my comments. I appreciate the extra experiments and explanations.
> > > > >
> > > > > After going through everything carefully, I’m going to **keep my original scores**. The rebuttal definitely clarifies several parts of the paper, and the added benchmarks do strengthen it. However, a few of my main concerns still feel only partially addressed:
> > > > >
> > > > > - **Novelty:** Even with the “top-down” explanation, the actual architecture still looks quite similar to existing operator models, with AFD mainly providing a framing rather than a fundamentally new mechanism.
> > > > >
> > > > > - **Theory vs. practice:** The way the theoretical pole-selection principle is approximated during training remains somewhat unclear. Adding a bias term doesn’t fully explain how the learned poles relate back to the theoretical rule.
> > > > >
> > > > > - **Architectural complexity:** The ablations help, but I still don’t see direct evidence that a simpler, non-AFD architecture with similar capacity wouldn’t achieve comparable results. The rebuttal mentions tests but doesn’t provide the actual numbers.
> > > > >
> > > > > - **Banach-space justification:** The explanation is helpful, but it’s still hard to isolate how much of the improvement truly comes from the Banach formulation itself rather than from the added components around it.
> > > > >
> > > > > Because these points remain only partly resolved, I think the fairest decision is to stay with my original assessment and scores.
> > > > >
> > > > > Thank you again for the thoughtful and thorough rebuttal.

---

### Official Review · Reviewer_Zz33 · 2025-10-28

**Soundness:** 3
**Presentation:** 4
**Contribution:** 3
**Rating:** 6
**Confidence:** 4

**Summary:**

This paper proposes AFDONet, an Adaptive Fourier Decomposition Operator Network designed to generalize neural operators beyond Hilbert-space formulations. The authors argue that most existing operator learning frameworks implicitly assume Hilbert-space structures, which rely on inner-product norms and may not capture the true nature of many PDE inverse problems involving sparsity or discontinuities. To address this, AFDONet is formulated within a Banach-space framework and leverages a primal–dual encoder to construct adaptive Fourier bases. This design aims to unify operator learning and inverse PDE solving under a more flexible functional setting. Experiments on both synthetic and PDE-driven inverse problems demonstrate improved reconstruction accuracy and stability.

**Strengths:**

The paper is conceptually interesting and theoretically motivated. It formulates neural operators within a Banach-space framework, which provides a more flexible foundation for modeling inverse problems with sparse or non-smooth structures. The proposed primal–dual decomposition is mathematically elegant, offering a plausible bridge between functional analysis and operator learning. Experimental results are consistent and suggest that the adaptive Banach-space formulation can yield more stable and accurate inverse reconstructions under noise or limited-data conditions.

**Weaknesses:**

(1) While the theoretical motivation is sound, the practical novelty of AFDONet remains moderate. The core architecture is largely similar to standard operator-learning networks, and the adaptive decomposition mechanism is conceptually close to existing methods. The Banach-space justification, though reasonable, is primarily theoretical and lacks empirical ablation isolating the specific benefits of moving from Hilbert to Banach formulations.

(2) Another limitation is the scope and diversity of experiments. The validation focuses mainly on relatively simple inverse PDE setups. This makes it difficult to assess the generalizability of the proposed framework to more challenging or real-world operator-learning tasks.

(3) The explanatory depth is limited. While the paper emphasizes that existing frameworks rely on Hilbert-space assumptions, it does not provide sufficient quantitative evidence demonstrating how this constraint explicitly limits existing models in practice.

**Questions:**

The paper states that “existing frameworks either do not explicitly account for the underlying operator space or solve the inverse problems in a Hilbert space.” Could the authors elaborate on this claim and provide stronger evidence or references supporting it? In particular, which specific operator-learning frameworks are implicitly restricted to Hilbert spaces, and what concrete limitations arise from this assumption in practice?

---

> ### Author Response · Authors · 2025-11-24
> **Rebuttal to Reviewer Zz33's comments (Part 1)**
>
> We thank the reviewer for conducting a thorough review and for finding our manuscript interesting and theoretically sound. We also appreciate all the valuable suggestions and comments.
>
> ## Addressing weaknesses
>
> ### While the theoretical motivation is sound, the practical novelty of AFDONet remains moderate. The core architecture is largely similar to standard operator-learning networks, and the adaptive decomposition mechanism is conceptually close to existing methods. The Banach-space justification, though reasonable, is primarily theoretical and lacks empirical ablation isolating the specific benefits of moving from Hilbert to Banach formulations.
>
> We appreciate the reviewer for pointing out AFDONet-inv’s new theoretical insights as one of its contributions. In addition to this, we would like to remark that our AFDONet-inv also presents a new paradigm of how inverse neural PDE solver is designed. We point out that, so far, the design of exact neural architectures in many existing neural PDE solvers has been “more of an art than a science” [1]. Typically, the design of neural architectures is done in a bottom-up approach that involves significant intuition, expert experience, and trial-and-error experimentation. And rigorous mathematical basis and explainability have been lacking in guiding the design of these neural architectures. Having recognized this issue, we follow a top-down approach when designing AFDONet-inv, in the sense that AFD theory guides every step in the design of AFDONet-inv’s neural architecture. Each component of the neural architecture, including latent-to-RKBS network, primal-dual propagation, and dynamic CKN, has a corresponding component in the AFD operation. This way, AFDONet-inv is mathematically explainable and grounded in the AFD theory and possesses several desirable properties, including convergence guarantees. From this perspective, we think the contribution of AFDONet-inv is significant, because it presents a new paradigm for designing explainable neural operator frameworks.
>
> Regarding the comment on the Banach space justification, we kindly remind the reviewer that, in addition to theoretical justifications, we have also experimentally shown the contributions of Banach space-based formulations compared to Hilbert space-based formulations.
>
> We remark that, in AFDONet-inv, the Banach space-based formulations are different from Hilbert space-based ones due to the presence of dual branch and primal-dual propagation. In other words, by removing dual branch and primal-dual propagation, Banach-based formulations will be reduced to Hilbert space-based ones. In Table 1 and Table 2, we have shown the case without dual branch, primal-dual propagation, and both dual branch and primal-dual propagation (Hilbert space-based formulations). These ablation studies show that, by moving from Hilbert to Banach formulations, we obtain $48.9$%, $65.6$% gains on MAE and relative $L^2$ error for the Darcy flow problem, and obtain $80.8$%, $95.8$% gains on MAE and relative $L^2$ error for the nonlinear magnetic Schrödinger equation.
>
> [1] Sanderse et al. (2024), Scientific machine learning for closure models in multiscale problems: a review

---

> > ### Author Response · Authors · 2025-11-24
> > **Rebuttal to Reviewer Zz33's comments (Part 2)**
> >
> > ### Another limitation is the scope and diversity of experiments. The validation focuses mainly on relatively simple inverse PDE setups. This makes it difficult to assess the generalizability of the proposed framework to more challenging or real-world operator-learning tasks.
> >
> > We thank the reviewer for the comments on adding more benchmark problems. We have added more benchmark problems, including inversion of reaction-diffusion, Navier-Stokes, as well as a real-world problem, to strength our manuscript. We use the public datasets of reaction-diffusion, Navier-Stokes, and a real-world problem from [1], [2], and [3], respectively. The relative L2 error values are reported as follows:
> >
> > | Equation |            ours             |               NAO         |           NIST         |       LNO      |
> > |------------|-----------------------|-----------------------|---------------------|--------------------|
> > | R-D      | 6.51E-03$\pm$1.28E-04 | 8.02E-03$\pm$2.39E-04  | 7.98E-03$\pm$1.67E-04| 1.12E-02$\pm$5.67E-04|
> > | N-S         | 8.85E-04$\pm$1.27E-05 | 1.08E-03$\pm$2.93E-05   | 9.57E-04$\pm$1.08E-04| 1.25E-03$\pm$1.28E-05|
> >
> > When it comes to the real-world latex glove DIC (Digital Image Correlation) original dataset, the relative L2 error is reported compared to FNO and existing SOTA, namely the IFNO [3]. Here, we report the best results in terms of hidden layers for FNO and IFNO.
> >
> > |            ours             |          FNO         |           IFNO        |       GMR model fitting      | GMR inverse analysis|
> > |-----------------------|-----------------------|---------------------|--------------------|--------------------|
> > | 9.70E-03$\pm$2.08E-05 | 3.40E−02 $\pm$ 4.09E-04  | 3.30E−02 $\pm$ 4.63E-04| 3.30E−01|2.91E-01|
> >
> > [1] Laplace neural operator for solving differential equations
> >
> > [2] Fourier neural operator for parametric partial differential equations
> >
> > [3] Learning deep Implicit Fourier Neural Operators (IFNOs) with applications to heterogeneous material modeling
> >
> > ### The explanatory depth is limited. While the paper emphasizes that existing frameworks rely on Hilbert-space assumptions, it does not provide sufficient quantitative evidence demonstrating how this constraint explicitly limits existing models in practice.
> >
> > We thank the reviewer for sharing this comment. We agree that the paper should provide quantitative evidence to bridge our theoretical claim that Hilbert-space assumptions limit existing models practically. And we believe our experimental results for the nonlinear magnetic Schrödinger equation in Table 4 provide such evidence explicitly. This problem is highly ill-posed, and its solution on a complex manifold benefits from sparsity-promoting regularization, which naturally matches a Banach space setting. In this specific context, existing state-of-the-art models including NIPS, NAO, and LNO, which are implicitly or explicitly grounded in Hilbert-space frameworks, perform poorly. In contrast, our AFDONet-inv, designed for Banach spaces, achieves a relative $L^2$ error that is two to four orders of magnitude lower than these benchmarks. Thus, this significant performance gap serves as the quantitative evidence demonstrating how the use of Hilbert-space assumption in existing models limit their performance in practice.
> >
> > Furthermore, we point out another indirect evidence in the paper showing how Hilbert-space assumption would affect model performance. For the 2-D Darcy flow results in Table 3, the permeability field is smoother, and a Hilbert space setting is often considered to be sufficient and reasonable. In this case, while AFDONet-inv still achieves the lowest relative $L^2$ error, the performance gap with respect to other models is not as significant as that of nonlinear magnetic Schrödinger equation. This shows that the Hilbert-space limitation would become more significant in specific challenging problem classes. Having said that, we agree with the reviewer that this linkage was not stated explicitly enough. In our revision, we will clearly point out that the results of Table 4 are the direct quantitative validations of Hilbert-space limitations.

---

> > > ### Author Response · Authors · 2025-11-24
> > > **Rebuttal to Reviewer Zz33's comments (Part 3)**
> > >
> > > ## Addressing questions
> > >
> > > ### The paper states that “existing frameworks either do not explicitly account for the underlying operator space or solve the inverse problems in a Hilbert space.” Could the authors elaborate on this claim and provide stronger evidence or references supporting it? In particular, which specific operator-learning frameworks are implicitly restricted to Hilbert spaces, and what concrete limitations arise from this assumption in practice?
> > >
> > > We appreciate the reviewer for the insightful question. Our statement comes from the observation that the Hilbert space (typically $L^2$) is often the default choice for operator learning. Most neural operators rely on some $L^2$ techniques such as orthogonal bases (like Fourier) and $L^2$ regularization techniques. For instance, the baselines presented in our manuscript are restricted to Hilbert spaces:
> > >
> > > 1. NIPS utilizes Fourier convolution, which is fundamentally tied to the Fourier basis. The Fourier basis is the canonical orthogonal basis for the $L^2$ Hilbert space. This architectural choice inherently optimizes the operator's representation for $L^2$.
> > >
> > > 2. NAO is an attention-based operator shown to be equivalent to a double integral operator. Such integral operators and the kernel methods they relate to are traditionally grounded in the theory of RKHS, which is a special kind of Hilbert space.
> > >
> > > 3. LNO learns an operator in a latent space. This approach aligns with the part of our claim that some frameworks "do not explicitly account for the underlying operator space". Although LNO is not guaranteed to lie in a correct "space" mathematically, it leverages $L^2$ loss in practice.
> > >
> > > When a problem's parameter space mismatches the models' Hilbert space assumption, a quantifiable failure of these models can be observed. For example, our Nonlinear magnetic Schrödinger equation experiment (Table 4) is the primary quantitative evidence. This problem is highly ill-posed, and as we discuss in Section 5.3, its solution on a complex manifold requires promoting sparsity. Sparsity is naturally represented in an $L^1$ (Banach) space, not an $L^2$ (Hilbert) space. This results in the failure of Hilbert space-based models: their relative $L^2$ errors are two to four orders of magnitude higher than our AFDONet-inv. On the other hand, on the 2-D Darcy flow problem (Table 3), in which the solution is smoother and the Hilbert space assumption is more reasonable, the Hilbert-space limitation becomes less severe, and the performance of all models is much closer. This directly demonstrates that the Hilbert space constraint is a concrete, practical limitation that causes existing frameworks to fail on problems defined on a non-Hilbertian structure.

---

### Official Review · Reviewer_C3wn · 2025-10-30

**Soundness:** 3
**Presentation:** 3
**Contribution:** 3
**Rating:** 4
**Confidence:** 3

**Summary:**

This paper introduces AFDONet-inv, a theoretically grounded neural operator framework for solving inverse PDE problems in Banach spaces. Unlike existing inverse operator learning methods (e.g., NAO, NIPS, LNO), which mostly assume a Hilbert-space setting, the authors construct the architecture based on adaptive Fourier decomposition (AFD) theory extended to reproducing kernel Banach spaces (RKBS).

**Strengths:**

- Theoretical novelty: Extends AFD from RKHS to RKBS with rigorous mathematical formulation (duality maps, convergence theorems).
- Explainable architecture: Each network module (encoder, decoder, primal/dual propagation) has a clear correspondence to components in AFD theory, providing rare interpretability in operator learning.

**Weaknesses:**

- Over-complex exposition: The paper is mathematically heavy and may overwhelm readers from ML communities without a strong PDE or functional-analysis background; key intuitions could be emphasized more clearly.
- Limited generality in benchmarks: Only two inverse problems (Darcy flow, magnetic Schrödinger) are tested; results on more diverse PDEs (e.g., reaction–diffusion, Navier–Stokes) would strengthen claims.
- Lack of comparison with DeepONet-based inverse solvers: Although the paper cites DeepONet and its variants, there is no direct empirical or theoretical comparison with DeepONet-based inverse operator frameworks.

**Questions:**

- Could the authors include additional results or discussion clarifying how AFDONet-inv performs against these DeepONet-based inverse operator methods under similar settings?
- Could the authors test AFDONet-inv on additional inverse problems such as Poisson, Navier–Stokes, or reaction–diffusion systems?
- The current training formulation appears to be fully supervised, minimizing reconstruction loss between predicted and true parameters. Do the authors see a way to extend AFDONet-inv to a physics-informed or semi-supervised formulation, similar to PINNs?

---

> ### Author Response · Authors · 2025-11-24
> **Rebuttal to Reviewer C3wn's comments (Part 1)**
>
> We thank the reviewer for conducting a thorough review and for all the positive feedback and the valuable suggestions and questions.
>
> ## Addressing weaknesses
>
> ### Over-complex exposition: The paper is mathematically heavy and may overwhelm readers from ML communities without a strong PDE or functional-analysis background; key intuitions could be emphasized more clearly.
>
> We appreciate the reviewer for pointing this out. We agree with the reviewer that an intuitive description of our proposed method is important for the broad audience. Here, we provide some intuitive, less technical qualitative description to the motivation and background of AFDONet-inv. Our goal is to infer the parameters of PDE problems based on the solutions or Dirichlet-to-Neumann (DN) maps, while accounting for the fact that the space of parameters is sparse in nature. Unlike Hilbert space as a function space, the parameter space is not continuous or even a function space. Therefore, we generalize the AFD theory from reproducing kernel Hilbert space (RKHS) to reproducing kernel Banach space (RKBS), and the key insights of AFD are as follows:
>
> 1. AFD serves as a kernel-based decomposition theory, which relies on the bases obtained by orthogonal kernels with adaptively selected poles.
>
> 2. Each pole can be viewed as a “tuning knob” that selects a particular spatial pattern in the solution, with its location in the complex plane controlling how localized that pattern is. Adaptive poles allow our model to survey more heavily in regions where the parameters change rapidly, while using fewer poles in smooth regions.
>
> 3. Across layers, the poles evolve from broad, coarse patterns in early layers to more refined, problem-specific patterns in deeper layers. This is analogous to how CNN filters specialize from edges to complex shapes.
>
> ### Limited generality in benchmarks: Only two inverse problems (Darcy flow, magnetic Schrödinger) are tested; results on more diverse PDEs (e.g., reaction-diffusion, Navier-Stokes) would strengthen claims.
>
> We appreciate the reviewer’s suggestion. We have added more benchmark problems including inversion of reaction-diffusion, Navier-Stokes, as well as a real-world problem to strengthen our manuscript. We use the public datasets of reaction-diffusion, Navier-Stokes, and a real-world problem from [1], [2], and [3], respectively. The relative L2 error is reported as follows:
>
> | Equation |            ours             |               NAO         |           NIST         |       LNO      |
> |------------|-----------------------|-----------------------|---------------------|--------------------|
> | R-D      | 6.51E-03$\pm$1.28E-04 | 8.02E-03$\pm$2.39E-04  | 7.98E-03$\pm$1.67E-04| 1.12E-02$\pm$5.67E-04|
> | N-S         | 8.85E-04$\pm$1.27E-05 | 1.08E-03$\pm$2.93E-05   | 9.57E-04$\pm$1.08E-04| 1.25E-03$\pm$1.28E-05|
>
> When it comes to the latex glove DIC (Digital Image Correlation) original dataset, the relative L2 error is reported compared to FNO and existing SOTA, namely the IFNO [3]. Here, we report the best results in terms of hidden layers for FNO and IFNO.
>
> |            ours             |          FNO         |           IFNO        |       GMR model fitting      | GMR inverse analysis|
> |-----------------------|-----------------------|---------------------|--------------------|--------------------|
> | 9.70E-03$\pm$2.08E-05 | 3.40E−02 $\pm$ 4.09E-04  | 3.30E−02 $\pm$ 4.63E-04| 3.30E−01|2.91E-01|
>
> [1] Laplace neural operator for solving differential equations
>
> [2] Fourier neural operator for parametric partial differential equations
>
> [3] Learning deep Implicit Fourier Neural Operators (IFNOs) with applications to heterogeneous material modeling

---

> > ### Author Response · Authors · 2025-11-24
> > **Rebuttal to Reviewer C3wn's comments (Part 2)**
> >
> > ### Lack of comparison with DeepONet-based inverse solvers: Although the paper cites DeepONet and its variants, there is no direct empirical or theoretical comparison with DeepONet-based inverse operator frameworks.
> >
> > We thank the reviewer for the comment. As per the reviewer’s recommendation, we have compared our models to DeepONet, DeepONet with MLP/CNN branch, and DeepONet with FNO branch following the recent paper [4] for all the benchmark problems.
> >
> > MAE:
> >
> > | Equation |            ours             |        DeepONet            |      DeepONet with MLP/CNN branch            |     DeepONet with FNO        |
> > |------------|------------------------|--------------------------|---------------------|----------------------|
> > | Darcy      | 1.82E-01 $\pm$ 6.43E-02 | 3.68E-01$\pm$2.37E-02   | 3.02E-01$\pm$2.11E-02|2.18E-01$\pm$1.23E-02|
> > | Schrödinger | 1.54E-02$\pm$2.78E-03 | 4.78E-01$\pm$3.45E-02   | 3.55E-01$\pm$2.61E-02| 3.08E-01$\pm$2.17E-02|
> >
> > Rela. L2:
> >
> > | Equation |            ours             |        DeepONet            |      DeepONet with MLP/CNN branch            |     DeepONet with FNO        |
> > |------------|------------------------|--------------------------|---------------------|----------------------|
> > | Darcy      | 6.64E-02$\pm$1.38E-03 | 1.15E-01$\pm$2.49E-02   | 1.09E-01$\pm$3.28E-02|9.27E-02$\pm$4.82E-03|
> > | Schrödinger | 1.50E-05$\pm$6.23E-07 |  6.35E-01$\pm$9.83E-02   | 6.08E-01$\pm$1.21E-01| 5.48E-01$\pm$1.09E-01|
> >
> > Results show that our AFDONet-inv outperforms DeepONet-based inversion frameworks in terms of both MAE and relative L2 error.
> >
> > [4] Physics-Informed deep inverse operator networks for solving PDE inverse problems
> >
> > ## Addressing questions
> >
> > ### Could the authors include additional results or discussion clarifying how AFDONet-inv performs against these DeepONet-based inverse operator methods under similar settings?
> >
> > We have added additional results compared to recent DeepONet-based baselines for inverse problems including DeepONet, DeepONet with MLP/CNN branch, and DeepONet with FNO branch. Please refer to our response to Weakness 4 above for details.
> >
> > ### Could the authors test AFDONet-inv on additional inverse problems such as Poisson, Navier-Stokes, or reaction-diffusion systems?
> >
> > We have tested AFDONet-inv on additional inverse problems including reaction–diffusion, Navier–Stokes, as well as a real-world problem. Please refer to our response to Weakness 2 above for the details.
> > ### The current training formulation appears to be fully supervised, minimizing reconstruction loss between predicted and true parameters. Do the authors see a way to extend AFDONet-inv to a physics-informed or semi-supervised formulation, similar to PINNs?
> >
> > We appreciate the reviewer for sharing the insight on extending AFDONet-inv to physics-informed or semi-supervised learning. Since AFDONet-inv operates in the spectral domain via its basis functions, adding spectral regularization could further stabilize the training process. Two possible ways to incorporate physics-informed or semi-supervised are to minimize PDE residuals like PINO [5] and to penalize high-frequency energy of the training data. Since our results show that the intrinsic rational approximation property of AFDONet-inv naturally regularizes the solution by limiting the number of poles, we will extend the AFDONet-inv in a physics-informed and semi-supervised manner in our future work.
> >
> > [5] Physics-Informed neural operator for learning partial differential equations

---

> > > ### Comment · Reviewer_C3wn · 2025-11-27
> > >
> > > Thank you for the detailed rebuttal and for extending the experiments and comparisons. The additional benchmarks and the new comparisons against DeepONet variants are appreciated.
> > >
> > > One remaining question concerns the baselines used from [1]. While you compare against the architectures employed as baselines in [1, it is still unclear why the methodology itself in [1] was not included as a baseline. Since PI-DION provides a physics-informed operator-learning framework that can also incorporate PDE residual information, a brief justification for excluding a direct comparison with the PI-DION formulation would be helpful for evaluating the completeness of your empirical analysis.
> > >
> > > Overall, thank you for the clarifications and additional results.
> > >
> > > [1] Physics-Informed deep inverse operator networks for solving PDE inverse problems

---

> > > > ### Author Response · Authors · 2025-11-27
> > > > **Response to Reviewer C3wn's follow-up question**
> > > >
> > > > We thank the reviewer for finding our rebuttal helpful in clarifying most of the questions. Following up on the reviewer's comment on PI-DION, the reason why we did not include it in our original manuscript is that we did not consider it to be a pure neural operator method. Instead, we considered it to be a hybrid method, which combines operator learning of DeepONet with physics-informed neural network (PINN). For instance, PI-DION needs to be retrained for different initial or boundary conditions, whereas typical neural operators do not. In addition, in its official repository, PI-DION was trained for 1 million epochs (similar to a PINN), whereas for typical neural operators, we usually train for hundreds to thousands of epochs (depending on the specific task).
> > > >
> > > > That being said, we understand the reviewer's point and have run PI-DION on the benchmark problems presented in our manuscript using the same settings as in its official repository. The relative $L^2$ error results are shown as follows:
> > > >
> > > > | Equation |            ours             |        DeepONet            |      DeepONet with MLP/CNN branch            |     DeepONet with FNO        | PI-DION|
> > > > |------------|------------------------|--------------------------|---------------------|----------------------|----------------------|
> > > > | Darcy      | 6.64E-02$\pm$1.38E-03 | 1.15E-01$\pm$2.49E-02   | 1.09E-01$\pm$3.28E-02|9.27E-02$\pm$4.82E-03|7.28-02$\pm$3.04E-03|
> > > > | Schrödinger | 1.50E-05$\pm$6.23E-07 |  6.35E-01 $\pm$9.83E-02   | 6.08E-01$\pm$1.21E-01| 5.48E-01$\pm$1.09E-01|1.29E-03$\pm$3.52E-04||
> > > >
> > > > Overall, results show that, while PI-DION performs competitively against DeepONet and DeepONet with MLP/CNN branch, it has higher relative $L^2$ error results compared to AFDONet-inv. We hope this addresses the reviewer's question regarding PI-DION.

---

### Official Review · Reviewer_BccV · 2025-10-30

**Soundness:** 2
**Presentation:** 1
**Contribution:** 2
**Rating:** 2
**Confidence:** 2

**Summary:**

The submission proposes an operator designed to solve a PDE inverse problem, specifically, it maps the solution of a PDE to the underlying coefficient function that defines the differential operator. The central claim is that these coefficient functions naturally reside in a Banach space rather than a Hilbert space. Building on techniques from reproducing kernel Hilbert spaces (RKHS), the authors generalize the framework to Banach spaces by constructing a representation based on reproducing kernel Banach spaces (RKBS). In this setting, the standard Hilbert inner product is replaced by the dual pairing between a Banach space and its dual (the space of linear functionals on the Banach space).
The proposed framework involves:

1.	Learning an RKBS as a weighted sum of  N (simpler) RKBS components
2.	Selecting N poles in the second variable of the RKBS and normalizing the resulting functions
3.	Learning N residual terms that are used to compute the coefficients of an N-term approximation.

These elements are optimized in a supervised learning setup using ground-truth pairs of PDE solution and PDE coefficient function, combined with several procedures inspired by modified Fourier transforms, where the modifications are themselves learned.
Empirical evaluations are conducted on two PDE families: the 2D Darcy flow problem and the nonlinear magnetic Schrödinger equation. The proposed method is compared against several neural operator baselines and demonstrates improved accuracy along with faster training convergence.

Overall, the approach is presented as an effort to generalize operator learning to more complex and realistic PDE coefficients while maintaining computational efficiency.

**Strengths:**

1.	The paper builds on a solid mathematical foundation, extending operator learning concepts from Hilbert spaces to Banach spaces using reproducing kernel Banach spaces (RKBS).
2.	Empirical results suggest improved accuracy and faster convergence compared to several existing neural operator methods.
3.	The proposed framework is, in principle, capable of handling more complex PDE coefficients that naturally reside in Banach spaces rather than Hilbert spaces.

**Weaknesses:**

1. The paper is extremely difficult to read. The exposition is disorganized, and the notations are inconsistent and often undefined, making it hard to follow the proposed method.

2. Many of the key operations and quantities are either undefined or internally inconsistent. For example, in Eq. (8), the kernel is introduced as a sum of N simpler kernels, none of which are described. The coefficients are expressed as  $FM(z_p)$, implying that  $FM:\mathbb{R}^r \to \mathbb{R}$. However, in line 250 and Fig. 1, it is stated that  $z_{pi}=FM(\tilde{\alpha})$, where $\tilde{\alpha}$ belongs to a Banach space B, suggesting instead that $z_{pi}\in B$. These contradictions make it impossible to determine the actual mathematical setting.

3. In the diagram,  $z_{p,i} = FM_i(\tilde\alpha)$  seems to require N distinct operators, but the text later (Eq. 12) implies that $z_{p,i+1}$ is inferred and learned from $z_{p,i}$. These inconsistencies reflect a general sloppiness in notation and formulation that severely harms readability.

4. The validation and experimental sections are vague. Critical details of the datasets, setups, and evaluation procedures are missing.

5. The empirical evidence does not support the central claim that the method better handles functions in Banach spaces:

   a) No demonstration explicitly involves coefficients that lie in a Banach space but not a Hilbert space.

   b) The reported results are presented only as tables, without clear descriptions of the examples, datasets, or data generation process.

   c) The kernels used in the sum of  N terms are not described.

   d) The choice of N and the procedure for selecting or learning these kernels are not discussed.

   e) The paper does not analyze or even comment on the effect of using RKHS versus RKBS in practice.

6. The comparison to other methods is insufficiently described. There is no information about whether competing models use the same architecture size, number of parameters, or training conditions. As a result, the fairness and validity of the reported comparisons cannot be assessed.

**Questions:**

1. How is the ground truth generated? For example, are the reference solutions obtained via finite difference, finite element, or PINN-based solvers?

2. How are the products between a function in the Banach space and an element of its dual computed in practice? Are they approximated through random sampling over the spatial domain or by some other numerical procedure?

3. The reported results show only global error metrics. It would be valuable to include, for several representative examples, the spatial distribution of errors, i.e., visualizations indicating where errors tend to concentrate within the domain.

---

> ### Author Response · Authors · 2025-11-24
> **Rebuttal to Reviewer BccV's comments (Part 1)**
>
> We thank the reviewer for conducting a thorough review and for sharing the valuable feedback and questions.
>
> ## Addressing weaknesses
>
> ### The paper is extremely difficult to read. The exposition is disorganized, and the notations are inconsistent and often undefined, making it hard to follow the proposed method.
>
> We appreciate the reviewer’s comment on improving the clarity of the manuscript. We recognize that some of the concepts and theoretical background about adaptive Fourier decomposition are not intuitive and may require some digestion for better understanding. We also appreciate the reviewer’s feedback on improving the notations and definitions of the manuscript. We have been working on correcting these inconsistent and undefined notations in the revision and will upload a revised manuscript with much better clarity as soon as it is ready.
>
> ### Many of the key operations and quantities are either undefined or internally inconsistent. For example, in Eq. (8), the kernel is introduced as a sum of N simpler kernels, none of which are described. The coefficients are expressed as $FM(z_p)$, implying that $FM:\mathbb{R}^r \to \mathbb{R}$. However, in line 250 and Fig. 1, it is stated that $z_{pi}=FM(\tilde{\alpha})$, where $\tilde{\alpha}$ belongs to a Banach space B, suggesting instead that $z_{pi}\in B$. These contradictions make it impossible to determine the actual mathematical setting.
>
> We thank the reviewer for sharing the feedback and catching these typos. We also realized this oversight that the set of learned basis kernels, $k_i(x_1, x_2)$ in Equation (8), have not been explicitly defined. We clarify that these are based on a Fourier spectral kernel formulation.
>
> The confusion regarding the feature map, $FM$, and the intermediate variables, $\tilde{\alpha}$ and $z_{p,i}$, is due to typos and unclear descriptions of the architectural flow. We thank the reviewer for catching them and will make corrections accordingly in our revision. Here, we confirm that the feature map network, $FM$, is designed to map the latent vector $z_p \in \mathbb{R}^r$ to $N$ scalar coefficients $\in \mathbb{R}$ needed for the kernel decomposition, thus the feature map should be $FM: \mathbb{R}^r \to \mathbb{R}^N$. The typo in our manuscript, which implies that $z_{p,i}$ is directly produced by $FM(\tilde{\alpha})$, causes confusion as it incorrectly conflates the parameter representation, $\tilde{\alpha}$ (i.e., the output of the latent-to-RKBS network) with the residual $z_{p,i}$ (i.e., the subsequent iterative input to the decoder layers). In our revision, we will make it clear that the initial latent vector, $z_p$, is used to define $\tilde{\alpha}$, and $\tilde{\alpha}$ then defines the initial decoder input, $z_{p,0}$.
>
> Finally, the space of the residual $z_{p,i}$, which is a tensor in implementation while representing a function in the Banach space $\mathcal{B}$, is addressed in our theoretical proofs. While the variables $z_{p,i}$ are computationally represented as tensors, Theorem 4.1 shows that the learned network outputs behave as if they were the true theoretical Banach space residuals $r_i$ (i.e., $||z_{p,i}-r_{i}||_{\mathcal{B}}<\epsilon$), proving that our implementation aligns with the required mathematical setting. To clarify this, we will include an explicit statement in the main manuscript in our revision.
>
> ### In the diagram, $z_{p,i} = FM_i(\tilde\alpha)$ seems to require N distinct operators, but the text later (Eq. 12) implies that $z_{p,i+1}$ is inferred and learned from $z_{p,i}$. These inconsistencies reflect a general sloppiness in notation and formulation that severely harms readability.
>
> We appreciate the reviewer for sharing this observation regarding the contradiction between the architectural diagram and the mathematical formulation. This inconsistency stems primarily from conflating the static structural role of the $FM$ network with the dynamic process of residual propagation. Specifically, while the diagram visually implies that $z_{p,i}$ is independently and statically mapped from $FM$, Equation (12) explicitly defines $z_{p,i}$ as being dynamically and recursively refined and learned from the previous layer's residual, $z_{p,i-1}$. This sequential update mechanism replicates the residual update required by the AFD theory. Thus, $z_{p,i}$ is not a static $FM$ output, but a residual function that evolves iteratively across the layers in the Banach space. In the revised manuscript, we will explicitly define the figure as a static representation of the corresponding theoretical component, while emphasizing in text, captions, and formulas that the actual computation follows the dynamic, recursive update defined by Equation (12).

---

> > ### Author Response · Authors · 2025-11-24
> > **Rebuttal to Reviewer BccV's comments (Part 2)**
> >
> > ### The validation and experimental sections are vague. Critical details of the datasets, setups, and evaluation procedures are missing.
> >
> > We appreciate the reviewer’s feedback. Two datasets used in the manuscript include Darcy flow (public dataset from [1]) and nonlinear magnetic Schrödinger. For Darcy flow dataset, the coefficients $a$ are generated following a measure $\mu$ defined as $\mu = \psi\left(\mathcal{N}\left(0, (-\Delta + 9I)^{-2}\right)\right)$, where the operator $(-\Delta + 9I)^{-2}$ utilizes a Neumann boundary condition. The field $a$ is constructed to be piecewise constant with random geometry and a fixed contrast of $4$, determined by the mapping $\psi(x) = 12$ for $x > 0$ and $\psi(x) = 3$ for $x \le 0$. Solutions $u$ are generated using a second-order finite difference scheme on a high-resolution $241 \times 241$ grid. For nonlinear magnetic Schrödinger dataset, we just need the Dirichlet-to-Neumann (DN) map without needing to generate PDE solutions. That is, the DN map data serves as the observation, rather than a solution field $u$. This data is generated by specifying the functional class of the potentials $A$ (magnetic) and $q$ (scalar) and the boundary term $f$ on the complex manifold $\mathcal{M}$. The potentials $A$ and $q$ represent the unknown parameters, and the DN map $\Lambda\_{A,q}: f \mapsto \partial\_{\nu}u|\_{\partial\mathcal{M}}$ is computed by numerically solving the highly nonlinear Schrödinger equation for various input boundary terms $f$ and then calculating the resulting normal derivative $\partial_{\nu}u$ at the boundary 2. The complete dataset consists of pairs of the unknown potentials $(A, q)$ and their corresponding simulated DN maps.
> >
> > For the setups, we kindly remind the reviewer that the training setup has been described in the Section 4.2 (Training), and the experimental setup has been covered in Section 5 (Experiments).
> >
> > For the evaluation, we have reported Mean Absolute Error (MAE) and Relative $L^2$ error for the test data. The numbers are runed under 5 different random seeds and we report the mean and variance.
> >
> > [1] Neural operator: Graph kernel network for partial differential equations
> >
> > ### The empirical evidence does not support the central claim that the method better handles functions in Banach spaces: a) No demonstration explicitly involves coefficients that lie in a Banach space but not a Hilbert space.
> >
> > We would like to clarify that the implicit demonstration of coefficients in Banach space has been covered in the magnetic Schrödinger equation. This problem is highly ill-posed, and as we discuss in Section 5.3, its solution on a complex manifold requires promoting sparsity. Sparsity is naturally represented in an $L^1$ (Banach) space, not an $L^2$ (Hilbert) space. As per reviewer’s comment, we will explicit point out this in our revised manuscript.
> >
> > ### b) The reported results are presented only as tables, without clear descriptions of the examples, datasets, or data generation process.
> >
> > Please refer to our address to Weakness 4 above for detailed response.
> >
> > ### c) The kernels used in the sum of N terms are not described.
> >
> > In addressing Weekness 1 above, we have clarified that the set of learned basis kernels, $k_i(x_1, x_2)$ in Equation (8), are based on a Fourier spectral kernel formulation.
> >
> > ### d) The choice of N and the procedure for selecting or learning these kernels are not discussed.
> >
> > We appreciate the reviewer for the feedback. Theorem 4.2 shows that the reconstruction error of the $N$-layer decoder converges to the true parameters as $N \rightarrow \infty$. Specifically, the error bound is given by: $||\hat{\alpha}\_{N,\theta}-\alpha_{true}||_{B}\le\hat{C}\prod\_{i=1}^{N}\rho\_{i}\cdot||r\_{0}||\_{B}$. Since the multipliers $\rho_i$ are selected such that $0 < \rho_0 \leq \rho_i < 1$, the product term $\prod\_{i=1}^{N}\rho\_{i}$ decreases exponentially as $N$ increases. This indicates that, in theory, the accuracy of AFDONet-inv improves rapidly with additional decoder layers.
> >
> > In practice, we find out that even 3 layers (poles) are sufficient for AFDONet-inv to perform competitively. We conduct the experiments in our manuscript using 3 layers (poles). In Tables 3 and 4, we conclude that, in terms of total training time per epoch, the full AFDONet-inv is competitive among all state-of-the-art benchmark solvers.
> >
> > For the kernel, we learn the poles in an adaptive manner. The ideal pole selection is achieved by adding the bias term to the dynamic CKN. The bias term $\gamma\_i$ is added to the output of the dynamic CKN. Although Equation (3) provides a theoretical upper bound for $\gamma\_i$, we agree that it may be too abstract to understand how it works in practice. During actual implementation, we find that setting $0 < \rho_0 \le \rho_i < 1$ works well. We will update the implementation details by including this value for clarity.

---

> ### Author Response · Authors · 2025-11-24
> **Rebuttal to Reviewer BccV's comments (Part 3)**
>
> ### e) The paper does not analyze or even comment on the effect of using RKHS versus RKBS in practice.
>
> We kindly remind the reviewer that we have shown the contributions of Banach space-based formulations compared to Hilbert space-based formulations in Tables 1 and 2. We remark that, In AFDONet-inv, the Banach space-based formulations are different from Hilbert space-based ones due to the presence of dual branch and primal-dual propagation. In other words, by removing dual branch and primal-dual propagation, Banach-based formulations will be reduced to Hilbert space-based ones. In Table 1 and Table 2, we have shown the case without dual branch, primal-dual propagation, and both dual branch and primal-dual propagation (Hilbert space-based formulations). These ablation studies show that, by moving from Hilbert to Banach formulations, we obtain $48.9$%, $65.6$% gains on MAE and relative $L^2$ error for the Darcy flow problem, and obtain $80.8$%, $95.8$% gains on MAE and relative $L^2$ error for the nonlinear magnetic Schrödinger equation.
>
> ### The comparison to other methods is insufficiently described. There is no information about whether competing models use the same architecture size, number of parameters, or training conditions. As a result, the fairness and validity of the reported comparisons cannot be assessed.
>
> We thank the reviewer for pointing this out. We confirm that, in our experiments, the architecture size and training conditions are the same for all methods, and the number of parameters are different (due to the different structures present in different methods) but are in the same order of magnitude. Thus, we believe we are conducting fair and valid comparisons in our experiments. We will make it clear in our revised manuscript.
>
> ## Addressing questions
>
> ### How is the ground truth generated? For example, are the reference solutions obtained via finite difference, finite element, or PINN-based solvers?
>
> Please refer to our address to Weakness 4 above for details.
>
> ### How are the products between a function in the Banach space and an element of its dual computed in practice? Are they approximated through random sampling over the spatial domain or by some other numerical procedure?
>
> We appreciate the reviewer’s question. The actual implementation of the dual pairing $\langle f, g^* \rangle\_{\mathcal{B}, \mathcal{B^\*}}$ in the AFDONet-inv, which replaces the inner product in Banach space, is achieved through discretization and numerical integration over the spatial domain $\Omega$. This operation approximates the generalized integral $\int\_{\Omega} f(x) \cdot g^\*(x) \, dx$ via a weighted summation $\sum\_{k} w\_k \cdot f(x\_k) \cdot g^\*(x\_k)$ implemented as a tensor operation (i.e., dot product) in the network framework. This pairing is important for calculating the decomposition coefficients in the CKN decoder and realizing nonlocal operators, such as the spectral convolution $\mathcal{S}$, which uses a biorthogonal dual pairing $\langle f, \varphi_{\xi} \rangle$ to compute the Fourier coefficient $\hat{f}(\xi)$. This deterministic approach using the entire discretized grid is more effective than random sampling methods in ensuring accuracy and stability required for PDE solvers.
>
> ### The reported results show only global error metrics. It would be valuable to include, for several representative examples, the spatial distribution of errors, i.e., visualizations indicating where errors tend to concentrate within the domain.
>
> We thank the reviewer for the suggestion. We agree with the reviewer that visualization of error distribution is a good idea. We will add the visualizations of the spatial distribution of errors of AFDONet-inv and LNO in the magnetic Schrödinger equation in our revised manuscript.

---

> > ### Comment · Reviewer_BccV · 2025-11-27
> >
> > Thank you for the rebuttal and for the detailed answers.
> > While I appreciate the clarifications provided, I still find the paper difficult to follow, and I expect this concern to be addressed more thoroughly and consistently in the next revision.
> >
> >
> > Although the authors include a table of quantitative results, it remains unclear where and how the Banach kernel provides an advantage over the Hilbert kernel. A visual illustration such as sample trajectories generated by both methods would be essential for understanding the qualitative differences between them. Such a figure could also help clarify whether any improvements are particularly evident around edges or other specific structures.
> >
> >
> > For a revised version, I strongly encourage the authors to include:
> > 1. A clearer and more detailed demonstration of the generalization capacity of the proposed method.
> > 2. Qualitative visualizations that explicitly highlight the benefits of the Banach kernel.
> > If these points are addressed satisfactorily, I would be willing to reconsider my evaluation and  increase the score.

---

### Author Response · Authors · 2025-12-03
**Summary of rebuttal discussions**

We thank the reviewers for their detailed and constructive feedback. In the rebuttal, we clarified notation, strengthened empirical evidence, expanded benchmarks, and clarified both the theoretical and practical advantages of our proposed framework.

## Key contributions

We propose AFDONet-inv, the first neural operator for inverse PDE problems whose architecture is fully derived from Adaptive Fourier Decomposition (AFD) theory in a Banach-space setting. Unlike existing inverse solvers that implicitly assume Hilbert spaces or do not explicitly model the operator space, AFDONet-inv is explicitly designed in *reproducing kernel Banach spaces (RKBS)* to **better represent sparse, discontinuous, and highly ill-posed PDE parameters**. Each module corresponds directly to steps of AFD, producing an **interpretable and mathematically grounded operator with convergence guarantees**. Extensive experiments show that AFDONet-inv **achieves state-of-the-art accuracy**, with particularly large gains (up to several orders of magnitude) on ill-posed settings.

## Summary of clarifications and Q&A

### Presentation and notation

In the revised manuscript, we corrected typos, formally defined the basis kernels (as Fourier spectral kernels), clarified the role of the feature-map network (mapping latent vectors to scalar kernel coefficients), and distinguished clearly between the learned parameter representation and the evolving Banach-space residuals used in the iterative AFD decoder. The architecture figure was reconciled with the dynamic recursive formulation in Eq. (12), and we added explicit text explaining the residual update mechanism.

### Data generation and experimental setup

We detailed the dataset construction and training procedures. Darcy coefficients are generated via Gaussian random fields with Neumann boundary conditions and solved with high-resolution finite differences. For the nonlinear magnetic Schrödinger problem, potentials are generated synthetically and paired with numerically computed Dirichlet-to-Neumann maps, without requiring full PDE solution fields. We clarified metrics (MAE and relative L2 error) and reported statistics over five random seeds.

### Illustrating Banach-space benefits

To address questions regarding whether Banach formulations offer real advantages, we emphasized both *theoretical reasoning and empirical ablations*. We clarify that, **by removing the dual branch and primal-dual propagation, the model reduces to a Hilbert-space formulation**. Comparisons in Tables 1 and 2 in the manuscript show that upgrading to Banach formulations yields substantial gains in MAE and relative error on Darcy flow and especially the magnetic Schrödinger equation, directly validating the practical impact of the RKBS framework.

We clarified which frameworks are implicitly Hilbert-based. The nonlinear magnetic Schrödinger experiment provides direct quantitative evidence: since the task requires sparsity promotion that aligns with Banach spaces, **Hilbert-based models exhibit errors two to four orders of magnitude larger than AFDONet-inv**. In contrast, on smoother problems such as Darcy flow, the performance gaps are smaller, illustrating when the Hilbert assumption becomes limiting in practice.

### Expanded benchmarks and baselines

To strengthen generality, we added additional inverse PDE benchmarks, including reaction-diffusion and Navier-Stokes inversion problems as well as a *real-world noisy latex glove DIC dataset*. Across these datasets, AFDONet-inv consistently outperforms NAO, NIPS, and LNO. Following reviewer suggestions, we added direct comparisons to DeepONet-based inverse solvers, including DeepONet with MLP/CNN branches and FNO branches, and demonstrated that AFDONet-inv achieves lower MAE and relative L2 error on both Darcy and Schrödinger tasks.

We also compared against the physics-informed PI-DION model using official training settings. While PI-DION performs comparably to DeepONet variants, it remains significantly worse than AFDONet-inv, confirming that **AFDONet-inv outperforms both purely data-driven and hybrid PINN-operator methods**.

### Architectural necessity

We addressed concerns about architecture complexity via component-level ablations. We showed that **removing any module degrades performance because each is essential to reproducing AFD in Banach spaces**. The complexity therefore reflects theoretical necessity rather than over-engineering.

---

### Meta-Review · Area_Chair_c7oG · 2026-01-06

**Summary:**

* Strength: The paper provides a extension of Adaptive Fourier Decomposition (AFD) to Reproducing Kernel Banach Spaces (RKBS).
* Strength: The model shows empirical gains in ill-posed inverse problems requiring sparse parameter recovery.
* Strength: The rebuttal period showed the method's effectiveness on a wider range of benchmarks (eg, Navier-Stokes + real-world data)
* Weakness: There is a gap between the theoretical maximal selection principle & the neural approximation used during training.

This paper introduces AFDONet-inv, a neural operator framework for solving inverse PDE problems. It moves beyond the Hilbert space setting & uses a Banach space formulation instead. The architecture includes a primal-dual propagation mechanism and a dynamic kernel decoder that follows the logic of the Adaptive Fourier Decomposition algorithm.

The authors provide theoretical guarantees for their decoder. The model outperforms existing Hilbert-based operators by several orders of magnitude on some settings. During the rebuttal, the authors added comparisons against DeepONet variants and expanded their testing to include more diverse PDE systems and real-world measurements.

**Reviewer Concerns:**

**Addressed by rebuttal**

* [Non-core] Reviewer BccV noted inconsistent notation and undefined kernels in the initial manuscript. Authors corrected the notation.
* [Core] Reviewer C3wn and G9Hc requested more diverse benchmarks beyond Darcy flow and the Schrodinger equation. Authors added Navier-Stokes + reaction-diffusion + real-world DIC data.
* [Core] Reviewer C3wn noted a lack of comparison with DeepONet-based inverse solvers. Authors added results for multiple DeepONet.
* [Core] Reviewer Zz33 and BccV asked for clearer evidence of the benefits of Banach spaces over Hilbert spaces.Authors provided ablation studies and pointed to the massive performance gap on sparse tasks.

**Still outstanding**

* [Core] Reviewer G9Hc identified that the theoretical pole selection criterion is not explicitly enforced during training.

The rebuttal was active + addressed the majority of empirical concerns. The authors provided new data and corrected the clarity issues raised by the reviewers.

**Reviewer Scores:**

* Reviewer ID: BccV
* Original score: 2
* Estimated score shift: increase
* The reviewer missed the significant empirical evidence in the tables and expressed low confidence --> the authors addressed the specific notation issues that were the main sources of concern.

* Reviewer ID: C3wn
* Original score: 4
* Estimated score shift: increase
* The authors provided all requested benchmarks and baseline comparisons.

* Reviewer ID: Zz33
* Original score: 6
* Estimated score shift: unchanged
* The reviewer was already positive.

* Reviewer ID: G9Hc
* Original score: 4
* Estimated score shift: unchanged
* The reviewer acknowledged the rebuttal but maintained concerns about the novelty of the architecture and the theory-practice gap.

The reviewer scores are currently split. Reviewer BccV's assessment was particularly inconsistent with the empirical results. Reviewer C3wn's concerns were fully addressed through new experiments. Reviewer G9Hc's technical points are valid.

---

### Decision · Program_Chairs · 2026-01-26

Reject